# Efficient removal of nanoplastics from industrial wastewater through synergetic electrophoretic deposition and particle-stabilized foam formation

Amna Abdeljaoued [1,2], Beatriz López Ruiz [1,2], Yikalo-Eyob Tecle[1], Marie Langner[1], Natalie Bonakdar [2], Gudrun Bleyer[2], Patrik Stenner[1] & Nicolas Vogel [2] ✉

Microplastic particles have been discovered in virtually all ecosystems worldwide, yet they may only represent the surface of a much larger issue. Nanoplastics, with dimensions well below $1\,\mu m$, pose an even greater environmental concern. Due to their size, they can infiltrate and disrupt individual cells within organisms, potentially exacerbating ecological impacts. Moreover, their minute dimensions present several hurdles for removal, setting them apart from microplastics. Here, we describe a process to remove colloidally stable nanoplastics from wastewater, which synergistically combines electrophoretic deposition and the formation of particle-stabilized foam. This approach capitalizes on localized changes in particle hydrophilicity induced by pH fluctuations resulting from water electrolysis at the electrode surface. By leveraging these pH shifts to enhance particle attachment to nascent bubbles proximal to the electrode, separation of colloidal particles from aqueous dispersions is achieved. Using poly(methyl methacrylate) (PMMA) colloidal particles as a model, we gain insights into the separation mechanisms, which are subsequently applied to alternative model systems with varying surface properties and materials, as well as to real-world industrial wastewaters from dispersion paints and PMMA fabrication processes. Our investigations demonstrate removal efficiencies surpassing 90%.

Plastic materials and products have become ubiquitous in nearly every facet of daily life due to their cost-effective manufacturing and desirable qualities such as lightweightness, durability, and mechanical resilience. However, the widespread adoption of plastics has led to significant challenges, as they often accumulate in landfills or the natural environment, where they progressively degrade into smaller microplastic particles[1]. The generation and distribution of microplastics and its effects on ecology and biology has been intensely investigated[2,3]. Research found that microplastic particles are omnipresent in our ecosystems, ranging from the deepest oceans[4] to the

[1]Particle Processing, Process Technology & Engineering, Evonik Operations GmbH, Rodenbacher Chaussee 4, 63457 Wolfgang, Germany. [2]Institute of Particle Technology, Friedrich-Alexander-Universität Erlangen-Nürnberg, Cauerstrasse 4, 91058 Erlangen, Germany. ✉e-mail: nicolas.vogel@fau.de

summit of Mount Everest[5] and the Antarctic snow[6], and that such particles have already entered our food chain[7]. These findings underline the need to remove microplastics from the environment. Despite this challenge, the micrometer-scale dimensions of the particles offer a clear strategy and quality control as they can be removed by conventional filtration or sedimentation and are observable by optical microscopy or even the naked eye[8,9].

Unlike microplastics, the fate of plastic debris at the nanoscale remains largely unknown. Nanoplastic particles are too minuscule to be observed, characterized, or quantified using optical microscopy, and they easily pass through conventional filter systems[10]. Furthermore, they exhibit Brownian motion and do not readily settle, posing challenges to their efficient characterization and purification in real environmental samples[11]. These obstacles have impeded the reliable assessment of the extent of the nanoplastic pollution issue.

This methodological gap[11,12] is especially severe as nanoplastics can be more problematic compared to their microscopic counterparts because of their bioactivity: Their small size increases the risk of cellular uptake[13,14] and translocation into biological tissue[15], from where they can efficiently enter food chains and human metabolism[16]. While some nanoplastics may agglomerate upon exposure to biological media due to the formation of a protein corona, thus allowing the separation with conventional microplastic methodologies[14,16,17], nanoplastic particles have nonetheless been found e.g., in tap water[18] or edible fruits and vegetables[19]. Agglomeration can also be induced on purpose by the manipulation of the dispersions by the addition of ionic or reactive additives, e.g., in electrocoagulation[20–23] or ozonation processes[24] to enable the use of conventional removal techniques established for larger microplastic particles[25]. Electroflotation removes hydrophobic particles by adsorption to gas bubbles rising through the dispersion, which are generated by water splitting reactions at electrodes at the bottom of the reactor[26,27]. As a relevant example in the context of plastic pollution, electroflotation has been successfully employed in wastewater generated from solvent-based automotive paints, where the plastic pollution forms during washing steps[26]. Another factor enhancing particle capture efficiency can be the release of multivalent ions when sacrificial electrodes are used. These ions destabilize colloidal systems, causing agglomeration and thus facilitate their capture by the bubbles[27–29]. However, the separation efficiency decreases with increasing particle load in the wastewater[26,30], and the removal of dispersed, colloidally stable particles is inefficient due to the low adsorption energy of such hydrophilic particles to the bubble surface[31,32]. Membrane filtration is yet another strategy to remove such small particles, but suffers from clogging and does not fully remove the particles from the aqueous phase[33]. A potential solution to the nanoplastic problems is to take advantage of their colloidal properties, i.e., to generate force fields in which these small particles can move, and to control their interactions, so that the particles can be efficiently collected. Recently, the use of magnetic nanoparticles as scavenger materials operating by providing attractive colloidal interactions with different nanoplastic particles has outlined an efficient approach along these lines[34].

While colloidal particles do not easily sediment, they readily move in electric fields. Electrophoretic deposition (EPD), where particles are brought in contact with an electrode to form a solid film, is widely used to deposit uniform ceramic, polymeric or composite coatings, with applications ranging from corrosion protection to biomaterial design[35,36]. In EPD, charged particles dispersed in a liquid medium are attracted and deposited on a conductive substrate of opposite charge upon application of an electric field[37,38]. The process is scalable, easy to operate and, by the use of aqueous dispersions, environmentally friendly. A key consideration in typical aqueous EPD processes is the prevention of water electrolysis[39]. The evolution of oxygen and hydrogen bubbles hinders the formation of dense and uniform films[39], and is therefore detrimental for the coating process.

A second fundamental property of colloidal particles is their ability to adsorb to liquid interfaces. This adsorption reduces the air- and oil/water interphase and can therefore lower the total free energy[31]. The attachment strength is a strong function of the contact angle[32], making it inherently dependent on the surface properties of the dispersed particles[40–44]. The spontaneous adsorption of colloidal particles to interfaces has long been used to form highly stable emulsions, termed Pickering emulsions[45,46], and foams, known as particle-stabilized foams[47,48]. Conventional, colloidally stable particles dispersed in an aqueous medium typically do not form such stable foams. To efficiently create particle-stabilized foams, the surface chemistry of colloidal particles needs to be carefully adjusted in line with the contact-angle dependent attachment strength, typically by controlling the pH[49–52] or by introducing tailored functional groups to the particle surface[48].

In our approach to separating nanoplastics from industrial wastewater, we harness the synergistic interplay between two distinctive properties of colloidal particles: their capability to migrate within electric fields and their propensity to generate particle-stabilized foams. We take advantage of the surface charges of nanoplastic particles, which are commonly present due to the addition of ionic stabilizers[53], charged comonomers[53], or proteins in biological media[14], and apply an electric field to deposit particles onto an electrode. Contrary to a conventional EPD process, we actively employ the electrochemical gas bubble formation as an additional mechanism for the efficient nanoplastic removal. Importantly, the water electrolysis causes changes in the local pH in the vicinities of the electrodes, affecting the surface properties of the nanoplastic particles close to the provided gas/water interface in the form of the nascent bubbles. Protonation of surface charges renders particles more hydrophobic, making them susceptible to the interfacial adsorption process and therefore enable the formation of a particle-stabilized foam. Thus, this process synergistically combines gas evolution as a typical limitation of aqueous EPD processes with the formation of a particle-stabilized foam enabled by the local changes in pH in the direct vicinity of the emerging gas bubble at the electrode surface. These processes collude to efficiently remove nanoplastic particles from an aqueous dispersion.

We investigate this synergistic interplay in the electrophoretic deposition and particle-stabilized foam formation process (ePhoam) in detail and elaborate on the removal mechanism using a well-defined nanoplastic system functionalized with carboxylic acid groups. Subsequently, we explore the versatility of the process by changing both material and surface properties of such model dispersions. Finally, we translate our findings to less defined industrial wastewater, with a primary focus on polymethylmethacrylate (PMMA) nanoplastics. PMMA is widely used as a transparent, shatter-resistant alternative to glass. The annual worldwide production of MMA was estimated to be around 3.9 million tons in 2020 bound to increase to 5.7 million tons by 2028[54]. On an industrial scale, PMMA is synthesized by emulsion polymerization, which forms colloidal particles with dimensions below 1 μm in an aqueous phase[53]. These particles are subsequently removed and processed into the desired material. However, this process generates vast amounts of wastewater that retains a significant amount of the polymeric particles. The wastewater must either be combusted at high energy cost and loss of valuable resources, or in a worst case scenario, released into the environment[55]. While exact data is difficult to obtain, the scale of such wastewater generation is reported to be 300 tons per day for Mexican manufacturers alone[55]. Besides the obvious loss of valuable raw material, the potential discharge of PMMA wastewater into soil and groundwater raises environmental concerns[55] and harmful effects on aquatic organisms, such as developmental malformations in early life stages of *Xenopus laevis* frogs[56].

## Results and discussion

### Mechanism of nanoplastic removal

We employed an electrolytic cell to investigate the nanoplastic separation process from aqueous environments (Fig. 1a). To elucidate the mechanistic details, we first used a PMMA model system with carboxylic acid surface functionalization, synthesized by surfactant-free emulsion polymerization, with an average particle size of 361 nm (Supplementary Fig. 1)[57]. In the ePhoam process, two distinct mechanisms collude to separate the PMMA nanoplastic particles from the aqueous phase. Firstly, EPD causes migration and deposition of the negatively charged particles to the anode, similar to its classical application in coating technologies[58] (Fig. 1a, b). Secondly, the particles adsorb to the interface of oxygen gas bubbles formed via electrochemical water splitting at the anode (Fig. 1a, c). As a result, a particle-stabilized foam is formed[59,60], which removes particles from the dispersion.

The key factor enabling the formation of this particle-stabilized foam is the local change in pH in the vicinity of the electrodes caused by the electrochemical water splitting reaction[61]. At the anode, the release of protons locally decreases the pH, while the production of hydroxide ions increases the pH at the cathode (Supplementary Fig. 2). Fig. 1d, e shows the local evolution of the pH of our model aqueous colloidal dispersion (1 wt.% nanoplastic) under the application of a continuous DC electric field, measured via an ion-sensitive field effect transistor pH meter. The bulk pH of 2.7, resulting from the carboxylic acid surface functionalization, decreased at the anode to pH ≈ 1.7 within a minute, while the cathode region experienced a pronounced increase in pH up to pH ≈ 9.5.

The local pH change critically affects the colloidal properties of the model PMMA nanoplastic dispersion. Fig. 1f shows the zeta potential as a function of the pH. Due to the carboxylate surface groups, the negative zeta potential decreased with decreasing pH and

eventually reached an isoelectric point at pH ≈ 1.5. The pH sensitivity is caused by the protonation of the carboxylate groups at low pH values, which reduces the surface charge density of the particles. This reduction in zeta potential simultaneously decreases the hydrophilicity of the colloidal particles, which is crucial for the formation of the particle-stabilized foam. It leads to a change of the contact angle the particles can assume at an air/water interface, making the interaction with the hydrophobic gas phase more favorable[42,62]. Thus, the total energy of the system is decreased[32] and the particles adsorb more efficiently onto the emerging gas bubbles, producing the particle-stabilized foam. This proposed mechanism corroborates with evidence in literature, where stable foams of negatively-charged particles are achieved only after addition of salt, cationic surfactant, or acids[51,52,63]. To support the proposed mechanism, Fig. 1g-i compares the capability to form particle-stabilized foams in our setup, by running the experiment without one of the vital contributors (gas, pH-change, particles). In the absence of particles, oxygen bubbles were created at the anode by water splitting, but simply burst when reaching the water surface (Fig. 1g). In the absence of an applied voltage (and thus, water splitting reactions and local pH change at the anode), gas bubbles can be externally added by bubbling oxygen into the particle dispersion, resembling the strategy of a flotation process. However, no foam formation was observed (Fig. 1h), which is attributed to the hydrophilic character of the colloidal dispersions prepared by emulsion polymerization. Consequently, destabilization of the particles must be ensured for foam formation. Only during water electrolysis, the pH reduction in the vicinity of the emergent oxygen bubble at the anode alters the surface properties of the particles in situ without the need of any additives, and increases their adhesion strength to the gas/water interface[31,32,42,62,64]. This local pH change is the key for the formation of the particle-stabilized foam at the anode (Fig. 1i). The foam, which can be effortlessly extracted from the dispersion (see Fig. 1a and

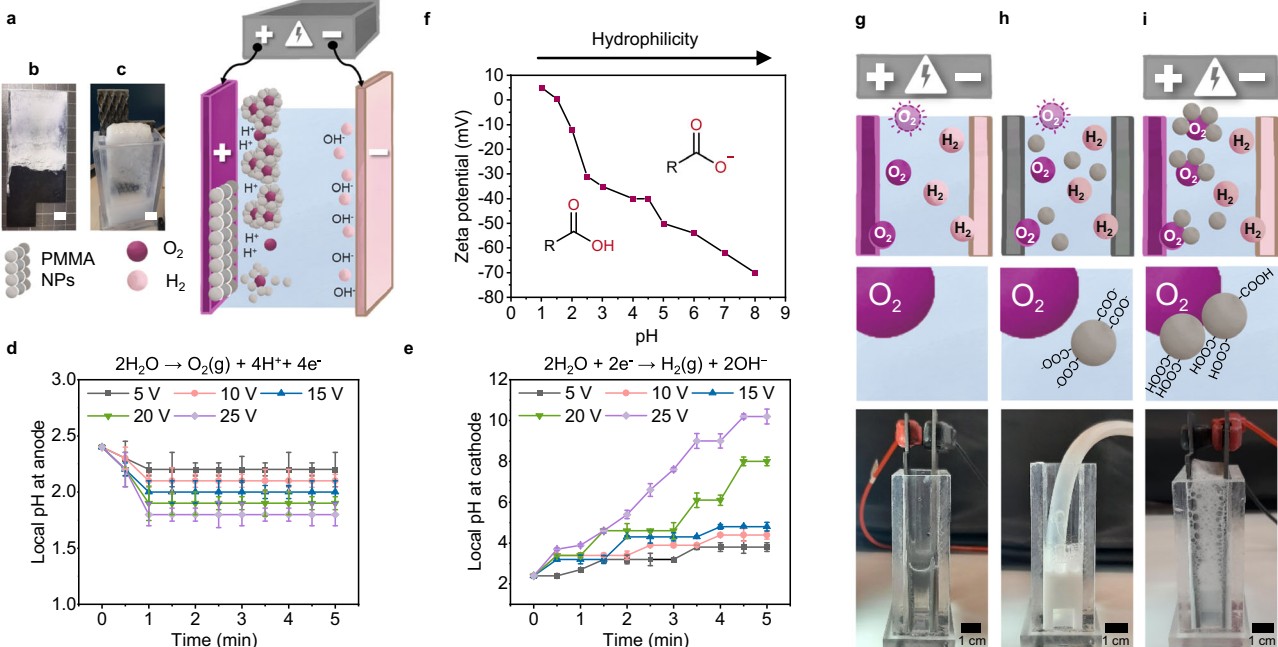

**Fig. 1 | Electrophoretic separation of nanoplastics from aqueous environments. a** Schematic illustration of the ePhoam process, which combines electrophoretic deposition (**b**), with the formation of particle-stabilized foam (**c**). Anode and O₂bubbles: purple; cathode and H₂bubbles: pink; PMMA nanoplastic particles: gray. Scale bars = 1 cm. **d** Local pH shift in the anode region, caused by the water oxidation reaction. Error bars represent standard deviation, $n = 3$ independent replicates. **e** Local pH shift in the cathode region caused by the water reduction reaction. Error bars represent standard deviation, $n = 3$ independent replicates. **f** Zeta

potential of the carboxylate-stabilized PMMA model system as a function of pH; (**g, h, i**) Fate of gas bubbles in the electrochemical setup. **g** In the absence of particles, gas bubbles are electrogenerated and are released to the environment. **h** Charged, hydrophilic PMMA particles do not efficiently adsorb to (artificially introduced) gas bubbles in the absence of an electric field. **i** The same particles adsorb to electrogenerated gas bubbles at the anode, as the local pH shift protonates the carboxylate surface groups, reduces their charge density and renders them more hydrophobic.

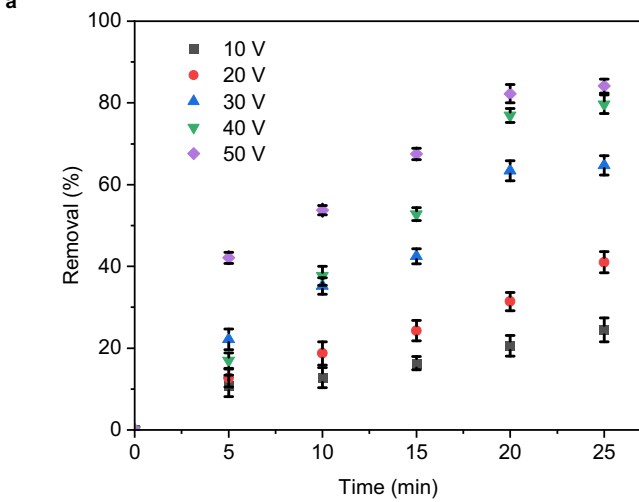

**a**

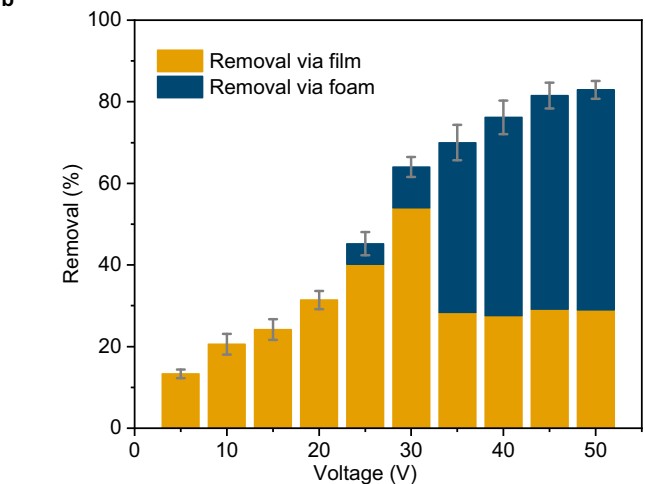

**b**

**Fig. 2 | Impact of process parameters on the removal of nanoplastics. a** Impact of operational time on the removal efficiency of the PMMA particles at different voltage; error bars represent standard deviation, $n = 3$ independent replicates. **b** Impact of the applied voltage on the removal efficiency in total and in the form of foam. Error bars represent standard deviation, $n = 3$ independent replicates.

Supplementary Fig. 3), remains stable for an extended period of time (Supplementary Movie 1). We further used mass spectrometry, evolved gas analysis (EGA) and selective blocking of individual electrodes to confirm that the stable foam indeed forms from oxygen bubbles at the anode (Supplementary Fig. 4). The lack of detected hydrogen confirms that hydrogen bubbles forming at the cathode do not contribute to particle-stabilized foam as the increased pH at the cathode does not reduce the absolute surface charge of our model particle system (Fig. 1f) and thus does not increase the propensity to form a stable foam. Unlike conventional electroflotation, the particle-stabilized foam forms directly at the electrode, allowing for the exploitation of both the phoretic movement of particles toward the nascent bubbles and of the local pH changes to enhance the interfacial adsorption efficiency. In contrast, in electroflotation, the adsorption process of particles to the bubble occurs within the dispersion volume and thus cannot exploit these effects. As a result, conventional electroflotation is less efficient in separating nanoplastics from colloidally stable dispersions as it does not generate a stable foam (see Supplementary Fig. 5 and Supplementary Movie 2).

Having established the twin-mechanism of the ePhoam process to separate particles from aqueous dispersions, we established parameters governing the efficiency of the process. For the study, we maintained a concentration of 1 wt.% of particles in the dispersion but note that separation can also be performed at lower concentrations (Supplementary Fig. 6). At fixed voltage and time, the removal efficiency increased with concentration, following Hamaker's equation which states that the mass of deposit at the electrode is directly proportional to the concentration of the charged moieties in the system[37]. In addition, at low initial concentrations, the mean path of a particle to reach the electrode significantly increases. Consequently, high removal rates for low concentrations require longer duration or higher voltages[38].

Supplementary Figure 7 shows that the removal efficiency was inversely proportional to the interelectrode distance, as predicted from the Ohmic potential, i.e., the loss of potential between the electrodes, which is proportional to the electrode distance. However, bubble formation caused a drop in the active electrode area as well as an increase in the electrolyte's ohmic resistance[64]. With decreasing distance, these factors dominate, preventing an efficient separation. As a compromise, an electrode distance of 1 cm was chosen.

Fig. 2a shows that for a fixed applied voltage, the removal efficiency linearly increased with time but eventually plateaued. This effect is expected in EPD with constant voltage, as the potential difference between the electrodes remains constant but the electric field influencing electrophoresis decreases with deposition time due to the formation of an insulating particle film on the electrode surface[37]. Fig. 2b summarizes the efficiency of the removal process as a function of the applied potential. With increasing voltage, we observed an increase in removal of particles in the form of a deposited film, as expected from the EPD process[37]. A potential of ≈ 25 V marks the onset of particle-stabilized foam formation. The bubble formation adversely affected the formation of particle films at the electrode, evidenced by a decrease in particle removal in the film state (Fig. 2b, yellow bars), and is therefore avoided in conventional EPD[39]. Nevertheless, this reduction is overcompensated by the formation of the particle-stabilized foam, which rapidly dominated the separation process at increasing voltages. This foam can be removed from the dispersion and thus contributes to the overall removal (Fig. 2b, blue bars). In total, the combination of EPD and particle-stabilized foams enabled us to remove more than 80% of all particles, with foam contributing over 50% of all removed particles. The foam amount and stability changed as a function of applied voltage (Supplementary Fig. 8). The foam volume increased with the voltage, as expected from the Faraday's law of electrolysis[65]. Above the threshold voltage for foam formation, all voltages produced a foam with long-term stability, although the initial dynamics changed. The highest voltage (50 V) showed the strongest decrease in foam volume before reaching a stable state in the measurement cell, despite yielding the largest foam volume. We hypothesize that this may be related to the increased bubble formation rate, possibly generating interfaces faster than the particles can attach to, and the increase in temperature, which lowers surface tension. In the following, we fixed the voltage to 50 V as this produced the highest total removal.

Next, we focused on the design of the electrodes, which is important in any electrochemical application as it influences kinetics and thermodynamics of electron transfer[66], process stability, and, in our case, may also influence the nucleation and growth of gas bubbles[67]. Besides the electrode material itself, the structure of the electrode can further influence these factors. Mesh electrodes, providing a higher surface area, are known to exhibit the lowest overpotential in hydrogen evolution reactions and lower heat generation compared to their flat analogues[68]. In Fig. 3, we compare a conventional stainless steel (St. steel) electrode used for the experiments shown above, with dimensionally stable anodes (DSA) (Supplementary Fig. 9). These electrodes are composed of a titanium substrate coated with iridium oxide ($IrO_2$), and are known to combine high catalytic activity and low overpotentials in oxygen evolution with high stability

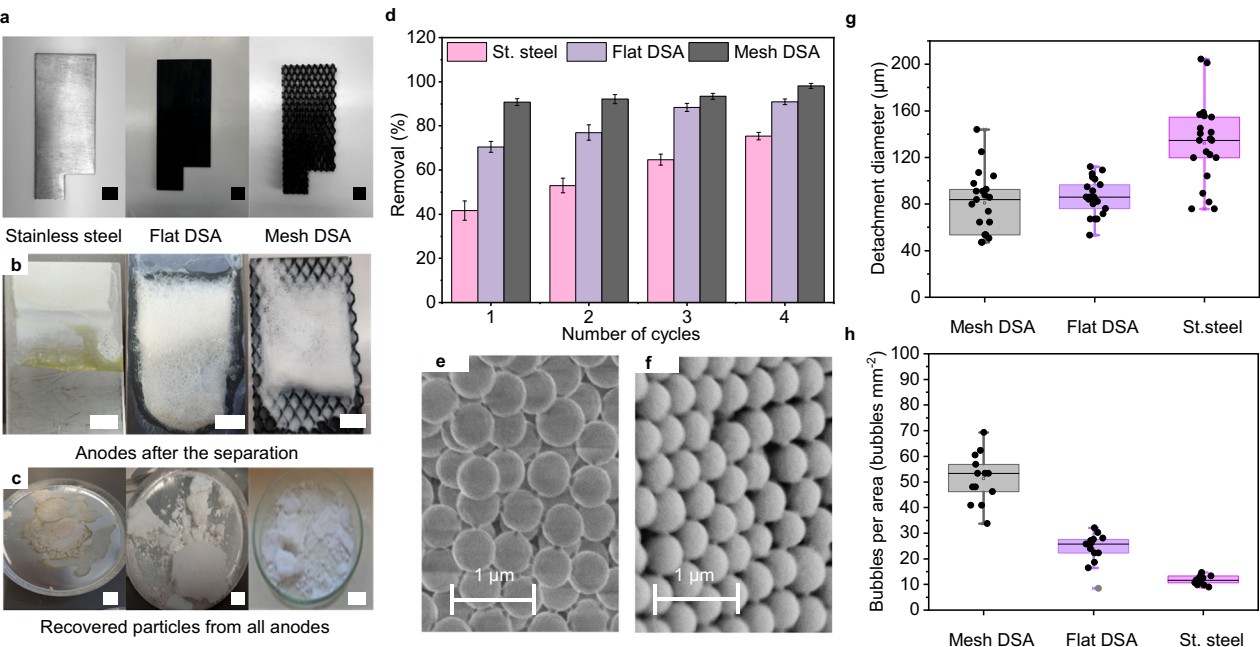

**Fig. 3 | Effect of electrodes for the separation process of nanoplastics.**
**a** Different electrodes tested for the separation of nanoplastics. **b** Anodes after the separation process. **c** Recovered nanoplastics from the different anodes.
**d** Comparison of the separation efficiency of the PMMA particles from the model system after 4 cycles of treatment using different electrode material and structure. Scale bars = 1 cm. Error bars represent standard deviation, $n = 3$ independent replicates. **e**, **f** SEM images of recovered film and foam, respectively. **g**, **h** Image analysis of the bubble formation process, observed via a high-speed camera on the different electrodes. **g** Detachment diameter of bubbles; **h** Number of bubbles per area. Box plots indicate median (middle line); 25th and 75th percentile (box); 5th and 95th percentile (whiskers); all data points (black points) and outliers (gray points).

against anodic corrosion and mechanical stability[1]. We further investigated the role of electrode geometry by using a DSA mesh electrode, with a calculated surface area that is approximately twice that of the flat electrode (Supplementary Fig. 10). As the DSA electrodes developed large amounts of foam in a short time, we resorted to a cyclic operation in which we assessed the particle removal by the ePhoam process as a function of 4 min operation cycles, where we removed the foam accumulated on the electrode surface after each cycle. This cyclic operation also ensured an efficient electrode performance, as particles deposited as a thin film on the electrode surface could be removed to avoid the formation of a passivating layer that reduces the electric field and thus the removal efficiency. Figure 3a, b shows photographs of the electrodes before and after one deposition cycle. The amount of foam generated by the DSA electrodes exceeded that of the stainless steel electrode. In addition, anode dissolution of the stainless steel electrode released $Fe^{3+}$ ions, evidenced by the yellow appearance of the recovered particles, which compromises both long term stability of the electrode and quality of the purified water phase. The superior performance of the DSA electrodes, which did not release any ions, shows that electrocoagulation[20–23], a phenomenon based on dispersion destabilization triggered by the release of multivalent ions, is not a dominant mechanism in our process.

Fig. 3d shows the cumulative yield of removed particles over four operation cycles. Using the mesh DSA electrode, 91% of nanoplastic particles were removed within the first cycles, compared to 70% for the flat DSA electrode and 42% for the stainless steel electrode. With increasing operation cycles, the removal efficiency increased in all cases, reaching up to 75% for stainless steel and 98% for the mesh DSA electrode. This setup thus left 2% of nanoplastic contamination in the aqueous phase, which meets the regulatory standards for the release of latex dispersions in the tested case of 1 wt.% initial concentration[69]. In addition, the removed material is available in the form of pure polymer powder composed of the original primary particles (Fig. 3c), as evidenced in SEM images of particle film and foam (Fig. 3e, f), and

can be readily re-integrated in the industrial PMMA production process.

We hypothesized that the enhanced efficiency of the mesh DSA electrode relates to the size of the emerging oxygen bubbles, which can more readily detach from the larger surface area and open structure of the electrode[67]. It is known from flotation processes that the separation efficiency of particulate materials greatly depend on bubble sizes[65,70] and that effective electroflotation is primarily attributed to the generation of small, uniform bubbles, which provide more surface area for particle attachment[70]. We investigated the size and formation rate of the oxygen bubbles on the different electrodes using a high-speed camera (Fig. 3g, h, Supplementary Movies 3–5).

Using image analysis of the movies, we extracted both the average bubble diameter at the time of detachment from the electrodes and the number of bubbles per area on the different electrodes (Fig. 3g, h, respectively). This analysis revealed that the bubbles released from the mesh DSA electrode had the smallest detachment diameter (81 μm), while bubbles released from the stainless steel electrode were much larger (135 μm) (Fig. 3g). In addition, the mesh DSA exhibited the highest number of bubbles per mm² (56 bubbles mm⁻²), which was twice the number observed on the flat DSA electrode (27 bubbles mm⁻²), and four times larger than the stainless steel electrode (13 bubbles mm⁻²) (Fig. 3h). Taken together, these data show that the mesh DSA electrode produces bubbles with the largest total gas/water surface area, leading to an enhanced capture and subsequent removal of the nanoplastic particles.

## Treatment of different model dispersions

We subjected a range of different model dispersions to the ePhoam process to probe the versatility of the method, in particular with respect to the material and the surface properties of the dispersions. To this end, we synthesized polystyrene (PS) colloidal particles with different surface functionalities by changing the comonomer in the surfactant-free emulsion polymerization process[71,72] (Fig. 4).

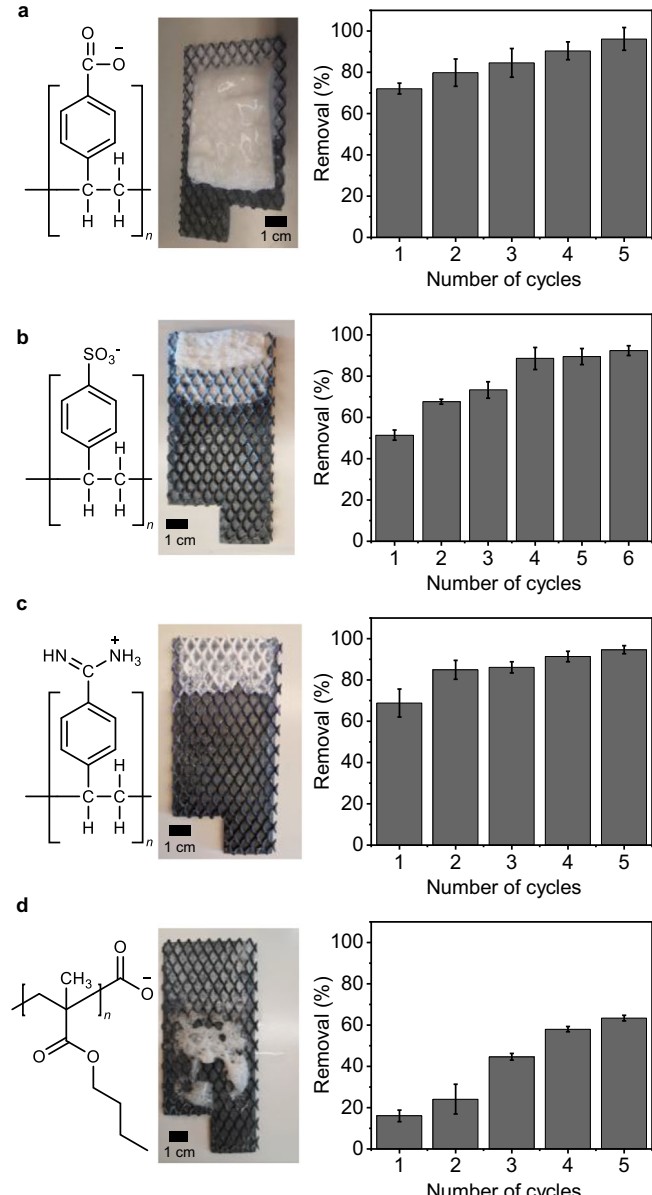

**Fig. 4 | Removal of model nanoplastic particles with different material and surface properties by the ePhoam process. a** PS-carboxylate particles; (**b**) PS-sulfonate particles; (**c**) PS-amidine particles, (**d**) PBMA particles. Error bars represent standard deviation, $n = 3$ independent replicates. Scale bars = 1 cm. All experiments were performed with mesh DSA electrodes.

Carboxylic acid-functionalized PS nanoplastic dispersions showed a similar behavior compared to the PMMA-based model system and exhibited removal efficiencies of 96% using 50 V and after 5 operation cycles, both by anodic deposit and particle-stabilized foam (Fig. 4a). Sulfonate-functionalized PS nanoplastic particles, which also have a negative surface charge (Supplementary Fig. 11), showed a comparable removal efficiency of 92% after 6 cycles (Fig. 4b). In this case, the initial removal at the first operation cycles was lower, and a higher voltage (70 V) was required. These differences may be caused by the more hydrophilic nature of the sulfonate group that requires higher proton concentration in the vicinity of the anode to efficiently lower the surface charges and form a particle-stabilized foam (Supplementary Fig. 11). Positively charged, amidine-functionalized PS nanoplastic particles also exhibited a pH-dependent change in surface properties, albeit with a decrease in surface charges toward higher pH values as

the amidine groups become neutralized by deprotonation (Supplementary Fig. 11). In this case, the separation achieved comparable yields, but the process was inverted, with the removal occurring at the cathode (70 V) in the form of deposited films and a particle-stabilized foam formed from the adsorption of particles to the evolving hydrogen bubbles enabled by the increased pH (Fig. 4c). Finally, we probed a third material class by synthesizing poly(butylmethacrylate) (PBMA) colloidal dispersions with carboxylic acid surface functionalities[73]. In this case, the surface properties are similar to the initial PMMA-based dispersion and both materials are methacrylates, yet differ in their glass transition temperature ($T_g$). While PMMA has a $T_g$ of $\approx 105\,°C$, the $T_g$ of PBMA is close to room temperature ($\approx 33\,°C$)[74], making the particles prone to film formation during operation. As a result, upon separation, this dispersion only formed an unstable foam and the deposit on the anode was rather transparent. Both observations relate to the low $T_g$, which causes the particles to sinter during the process, leading to a homogeneous film on the anode and a mechanically less stable foam. Consequently, the removal efficiency was lower and reached 63% after 5 cycles (Fig. 4d). Noteworthily, this dispersion had a higher conductivity, and thus required a lower voltage (32 V) for particle removal. These experiments demonstrate that the separation process can be applied to a range of different materials and surface properties, as long as the process conditions are adapted to the specific characteristics (surface charge, $T_g$, conductivity).

## Transfer to industrial wastewater

Having established the efficiency of the ePhoam process for various model colloidal dispersions, we applied the same principle to industrial wastewater, obtained from the emulsion polymerization-based PMMA production process. The solid content of this wastewater fluctuated between 5 and 7 wt.% and was diluted to 1 wt.% for reproducibility and comparison with the model system. Importantly, the zeta potential of the wastewater dispersion exhibited a similar decrease with decreasing pH values as the model dispersion of carboxylate-stabilized colloidal particles (Supplementary Fig. 12), indicating that it would be susceptible to the formation of a stable particle-stabilized foam. Figure 5a shows the efficiency of the PMMA nanoplastic removal from the industrial wastewater as a function of operation cycles. While the initial separation efficiency after the first cycle was lower compared to the model system, subsequent operation cycles increased the removal up to 95%. Photographs of the wastewater dispersions before and after the process underline the efficiency of the ePhoam process. The initially white dispersion (Fig. 5b) turned into a transparent liquid (Fig. 5c–e), which had a yellow color for the sample purified with a stainless steel electrode due to release of ferric ions, but remained completely colorless in case of the DSA electrodes (Fig. 5d, e).

To assess the environmental impact, we analyzed the chemical oxygen demand (COD) and the total organic carbon (TOC) of industrial wastewater before and after treatment (Fig. 5f). The TOC in the diluted wastewater (1 wt.%) decreased from 10,411 $mg\,L^{-1}$ before treatment to 215 and 208 $mg\,L^{-1}$ after treatment using flat DSA and mesh DSA, respectively. The COD of the diluted wastewater (1 wt.%) before treatment was 32,368 $mg\,L^{-1}$ and decreased to 695 (flat DSA) and 644 $mg\,L^{-1}$ (mesh DSA) after treatment. Thus, more than 97% of the organic compounds were removed. These values are close to the regulatory threshold values determined by the German Federal Ministry for the Environment, Nature Conservation, Nuclear Safety, and Consumer Protection (BMUV) for the release of natural rubber dispersions into the environment[2] (threshold COD = 150 $mg\,L^{-1}$). If the initial industrial PMMA dispersion is considered as wastewater from the chemical industry, the treated wastewater even falls within the regulations to be discharged in water bodies, according to these regulations[63].

SEM images and photographs of the particles recovered from the model dispersions (Fig. 3) already indicated that the particles remain

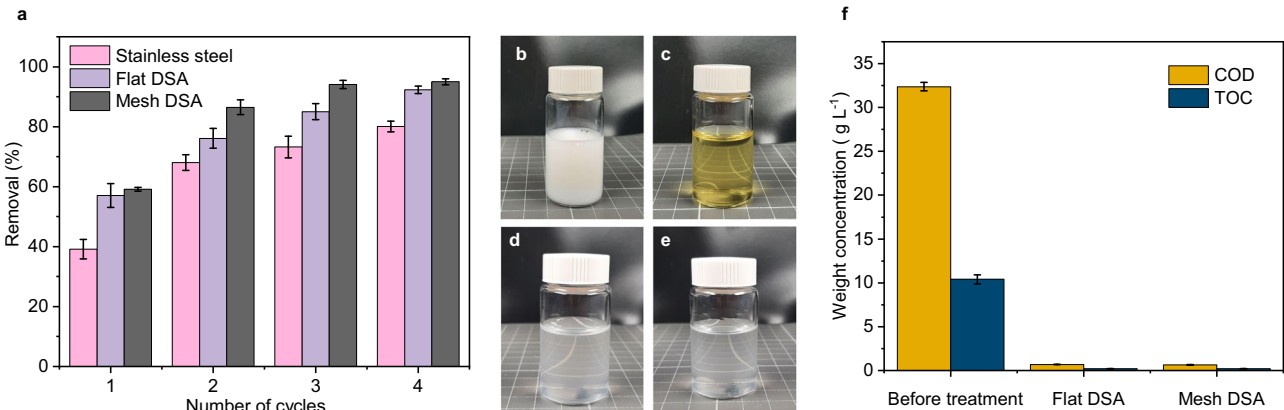

**Fig. 5 | Separation of nanoplastics from industrial PMMA wastewater.**
**a** Efficiency of PMMA particle removal during four cycles of treatment with different electrodes. Error bars represent standard deviation, $n = 3$ independent replicates. Photographs of the wastewater before (**b**) and after electrophoretic deposition/ particle stabilized foam formation process (**c**–**e**) using respectively stainless steel, flat DSA and mesh DSA. **f** Chemical oxygen demand (COD) and total organic carbon (TOC) analysis of industrial wastewater at 1 wt.% before and after treatment with flat and mesh DSA. Error bars represent standard deviation, $n = 3$ independent replicates.

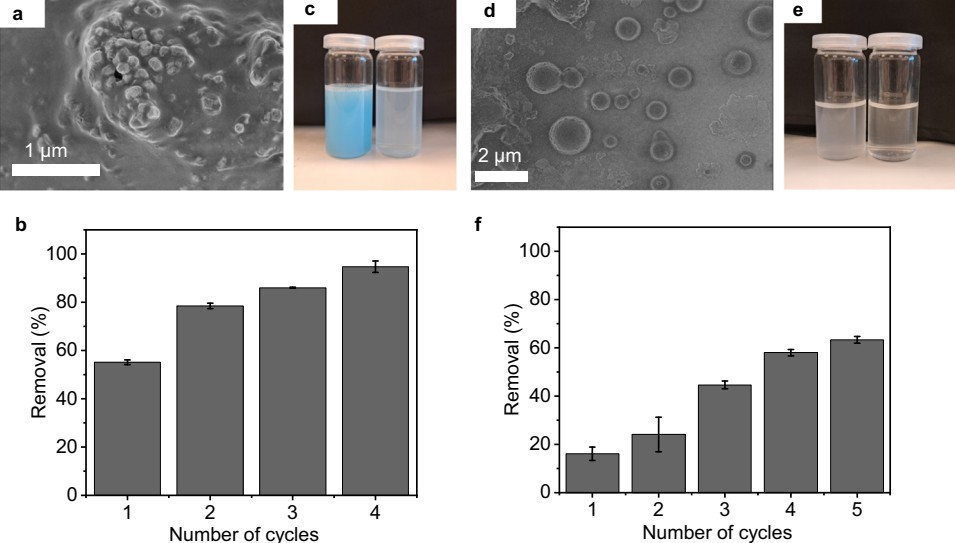

**Fig. 6 | Application of the ePhoam process to wastewater arising from different industrial products and processes.** SEM images of the particles, the wastewater dispersions before and after treatment, and the treatment efficiency as a number of treatment cycles: (**a**–**c**). **a**–**c** Wastewater from blue dispersion paint. **d**–**f** Wastewater from industrial eyeglass and contact lenses polishing processes. Error bars represent standard deviation, $n = 3$ independent replicates. All experiments were performed with mesh DSA electrodes.

intact and clean after separation and may thus be recycled in the industrial production process. To further confirm this recycling potential, we investigated the thermal properties, molecular weight distributions, chemical composition, as well as surface properties of the PMMA powder before and after separation (Supplementary Figs. 13–16, Supplementary Tables 7, 8). All measurements indicate that the separation process does not degrade or chemically alter the polymers, and thus indicate the possibility to recycle the recovered material in the production process for PMMA products.

Compared to other electrochemical-driven technologies (Supplementary Table 3) and incineration, which is the current technological treatment of industrial wastewater from emulsion polymerization-based processes, the ePhoam process is economically viable and cheap, in particular when including the ability to recycle the polymer. A direct comparison to incineration, using energy costs of July 2023, factoring in estimated operational- capital costs and payback for recycled polymer, and assuming a continuous process

(see below) shows a cost reduction of 89% and savings of ≈ 178 € per ton of wastewater (Supplementary Fig. 17). In addition, adoption of the ePhoam process in wastewater treatment at such industrial emulsion polymerization facilities would lower the carbon footprint by approximately 77% compared to the state-of-the-art waste treatment of incineration (details in Supplementary Notes 2,3 and Supplementary Fig. 17).

We investigate the versatility of the technology by subjecting wastewater from different industrial applications and processes to the ePhoam process. Figure 6a–c shows the removal of nanoplastic particles from a commercial dispersion paint. Dispersion paints are water-based materials that contain polymeric binder particles, typically with a size between 80 nm to 1000 nm[75], and inorganic pigment particles[76]. They are ubiquitously used as indoor wall paints and are applied in volumes of approx. 17 million tons per year[77]. When disposed inadequately, such paints form a large source of nanoplastic particles that are released into the environment. As an example, estimates for the

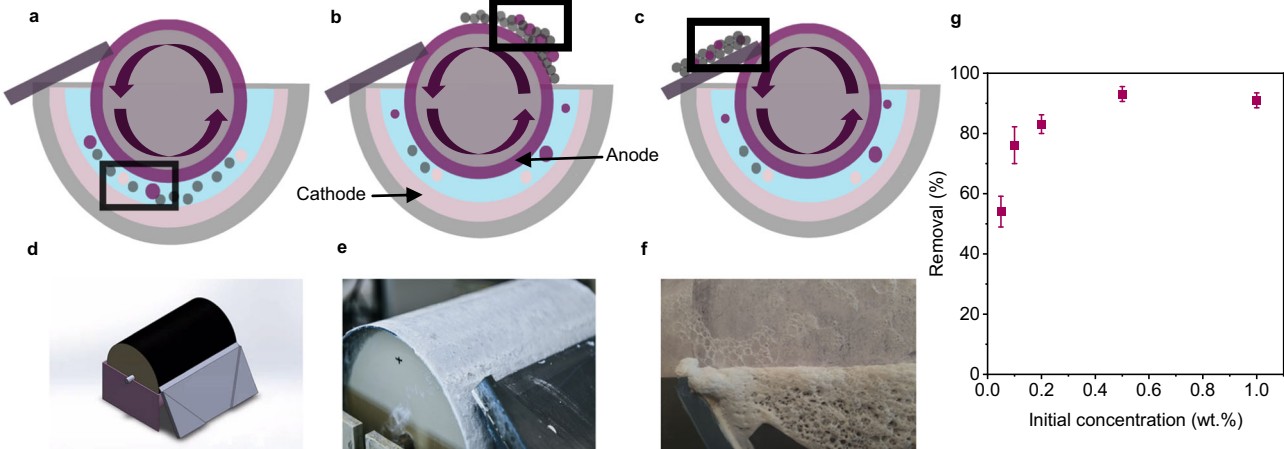

**Fig. 7 | Continuous operation and scale up, implemented by a "roller" pilot plant based on a rotating anode. a–c** Schematic illustration of the separation process. **a** A rotating anode attracts the nanoplastic particles from an aqueous dispersion. **b** The deposits are subsequently transported out of the water phase and collected by a doctor blade. **c** The replenished electrode re-immersed into the dispersion for a continuous process. Anode and O₂: purple; cathode and H₂: pink; nanoparticles: gray. **d–f** 3D illustration (**d**) and photographs of the plant during operation, showing the doctor blade (**e**) and the removal of particle-stabilized foam (**f**). **g** Impact of initial concentration on the removal efficiency using the roller system. Error bars represent standard deviation, $n = 3$ independent replicates.

Netherlands indicate a release of approx. 16 tons of plastic particles from washing out paint brushes[78]. Ultimately, the released particles find their way into aquatic ecosystems and may enter the food chain[79]. The challenge in separating such dispersion paint-based nanoplastics is the nature of the binder component itself, which is designed for efficient film formation[75]. This property, typically associated with a low glass transition temperature, can prevent the formation of particle-stabilized foams, as shown above for the PBMA model colloid. Nevertheless, the separation process was successful in removing approximately 92% of particles from the dispersion, despite the challenges posed by the multi-component system, the presence of molecular additives (including antimicrobial additives, as specified by the manufacturer), and the chemical nature of the binder (see Fig. 6c). The decreased particle content can be directly observed in photographs of the dispersion after the separation process by the absence of color and the reduced opacity (Fig. 6b).

Finally, Fig. 6d–f shows the results of the removal of micro- and nanoparticles from the wastewater of industrial eye glass and contact lens polishing, obtained from a company focusing on optical products. During this process, polymeric materials are released from the product to be polished in the form of micro- and nanoparticles with ill-defined shapes[80,81]. This wastewater thus constitutes the most heterogeneous system to be purified. SEM analysis confirmed the heterogeneous nature and showed the presence of fragments, fibers, and spherical particles with a broad size distribution and a large population around 300 nm – 70 nm (Supplementary Fig. 18). The materials composition was heterogeneous as well and included polycarbonate (PC), poly(allyl-diglycol carbonate) (PADC), polyurethane (PU), as well as thiourethane-based additives, as shown in Supplementary Figs. 19–21 using Fourier-transform infrared (FTIR) spectroscopy, pyrolysis-gas chromatography mass spectroscopy, and thermodesorption-gas chromatography-mass spectrometry, respectively. Figure 5f shows that even in this least-defined wastewater, approx. 65% of all particles were removed. We identify the low initial concentration of solids within the wastewater (0.03 wt.%) as the primary reason for this comparably low removal efficiency. Fewer particles are available to stabilize the bubble interfaces, rendering the formation of particle-stabilized foam less efficient[82]. While these results are in line with recent results for an oil-based flotation process[81] and our own investigation on the concentration-dependent separation efficiency (Supplementary Fig. 6), they underline the effectiveness of the ePhoam process in achieving significant reduction of nanoplastic content even under the least defined and dilute conditions.

## Toward a scalable process

While the multi-step processes using the transparent electrolytic cell shown in Fig. 1 enabled parameter studies and the fundamental elucidation of the mechanisms underpinning the ePhoam process, it lacks the scalability and throughput required for realistic treatment of industrial wastewater. In a proof-of-principle demonstration, we transferred the concept to a scaled pilot reactor capable of a continuous operation. In this "roller" setup, the anode is implemented as a DSA cylinder that continuously rotates against a semicircular, static cathode (Fig. 7a–c). The wastewater can be inserted on one side and removed on the opposite side of the reactor. During operation, the anode rotates, picks up particle deposits, both as foam and thin film, which is subsequently removed from the electrode by a doctor blade. As a result, the removed particles can be recycled, while the electrode surface is recovered and re-immersed (Supplementary Movie 6). Fig. 7d–f shows photographs of the setup during operation. In this pilot plant, 2.5 L total dispersion can be treated at a time. The removal efficiency of nanoplastic particles depends on the initial concentration, similar to the model system (Supplementary Fig. 6), and reaches values above 90% for concentrations of 0.05 wt.% and above, using a 2-hour process with 50 V as an applied voltage and 4.25 mA cm⁻² as a current density (Fig. 7g).

In summary, we demonstrate that the ePhoam process can be used to efficiently remove colloidally stable nanoplastic particles from industrial wastewater. Our investigations reveal that two mechanisms collude to provide this efficient removal. Electrophoretic deposition causes particle aggregation at the electrode surface, while gas bubble evolution continuously creates new gas/water interfaces onto which particles can attach. A reduction of local pH at the anode destabilizes the particles and renders them more hydrophobic, which increases the adsorption efficiency. As a result, a persistent particle-stabilized foam removes the particles from the water phase. With optimized conditions and a meshed electrode design to enhance surface area and bubble formation efficiency, a total of 95% of particles were successfully removed from industrial wastewater of the PMMA production and made available as a pure polymer powder to be reintegrated into the production process. The process is scalable and can be envisioned to be integrated into industrial wastewater treatment to efficiently remove nanoplastics and to recover valuable raw materials. The

combination of cost- and resource-efficient treatment, continuous operation, and full recyclability without degradation of the polymer material are important criteria for an industrial integration. Since the separation process is not specific to a certain material, it can be translated to wastewater from different industrial processes, from industrial polishing and cleaning solutions to dispersion-based paints and coatings. A further perspective application with a large environmental impact is the treatment of wastewater from the natural latex industry[83]. A future transfer to nanoplastic detection, analysis, and removal in environmental water can be conceptually envisioned, but necessitates fundamental research and technological developments on understanding the role of protein adsorption and aging of such particle systems in biological environments and on the process efficiency with minute particle concentrations.

## Methods

### Materials

Methyl methacrylate (MMA, ≥99%, extra pure), sodium hydroxide (≥98%) and aluminum oxide 90 neutral ($Al_2O_3$) were purchased from Carl Roth. Styrene (ReagentPlus®, ≥99%), butyl methacrylate (BMA, 99%), acrylic acid (AA, 99%), sodium 4-vinylbenzenesulfonate (≥90%), (vinylbenzyl)trimethylammonium chloride (99%), 2,2-azobis(2-methyl-propionamidine) dihydrochloride (97%) and ammonium persulfate (APS, ≥98%) were purchased from Sigma-Aldrich. Nitric acid (65%) and calcium sulfate dihydrate were purchased from Merck. For water, a Purelabflex (elga Lab water) purification unit was used (18.2 MΩ cm).

### Emulsion polymerization of model nanoplastic dispersions

PMMA colloidal particles were synthesized by surfactant-free emulsion polymerization. Bidistilled water was heated to 90 °C in a round-bottom flask equipped with a reflux condenser under stirring at 500 rpm and constant purging with $N_2$. Subsequently, the monomer MMA was added to the heated water through a syringe. Before addition, the monomer was purified to remove the inhibitor by extraction with 10 wt.% NaOH and column chromatography with neutral $Al_2O_3$. 10 min after the addition of the monomer, the comonomer AA - dissolved in 10 g of bidistilled water–was added to the reaction mixture. After an additional 10 min, the initiator APS was added. The reaction was allowed to proceed for 12 h and stopped by cooling the reaction mixture to room temperature. The resulting polymer colloidal dispersion was filtered through a lint-free tissue (VWR).

Similarly, PS and PBMA colloidal dispersions were synthesized using surfactant-free emulsion polymerization. All the syntheses details are given in the Supplementary Note 4. Before use, the styrene and BMA were washed with 10 wt.% NaOH aqueous solution to remove the inhibitor and then passed through an $Al_2O_3$ column to dry. The purified monomers were stored no longer than 2 months at 8 °C before use. At the end, to separate any coarse fractions, the final colloid was filtered through a KIM Wipe (Kimberly-Clark).

### Characterization of colloidal dispersions and recovered particles

Scanning electron microscopy (SEM, Gemini SEM 500, Zeiss) using an acceleration voltage of 1 kV and the SE2 detector; and dynamic light scattering (DLS) (Zetasizer Pro Blue ZS, Malvern Panalytical Ltd) were used to analyze the diameter of PMMA, PBMA, and different PS particles. The particle dispersions were diluted to 0.0001 wt.% in bidistilled water and ultrasonicated for 2 min to avoid agglomeration and equilibrated to 25 °C for 2 min before the size measurement. The z-average and dispersity were used to characterize the particle size distribution of particles. The size measurements were conducted at least five times and average values were calculated. The SEM characterization results showed that the model particles used in this experiment were spherical, uniformly dispersed and had no obvious particle agglomeration.

The results of DLS analysis showed that the average particle size of PMMA particles was 361 nm, an average zeta potential of −35 mV (pH 2.4). The average particle size of PS-carboxylate; PS-sulfonate; PS-amidine and PBMA-carboxylate particles were 295 nm; 137 nm; 303 nm; and 221 nm, respectively. The average zeta potentials were −49 mV; −55.7 mV; 32.1 mV; and −52 mV, respectively. The dispersity for all samples was below 5%.

Eyeglass and contact lenses polishing wastewater was characterized using different analytical tools. The IR spectroscopy was measured in transmission on the Continuum microscope (Thermo Fisher Scientific) coupled to a Thermo Fisher Scientific iS50 FT-IR spectrometer. The samples were dried (16 h, 90 °C), applied to KBr pellets, and measured in transmission using an IR microscope. Pyrolysis gas chromatography mass spectrometry (GC-MS) was measured using a QP2010plus / GC2010 with Pyrolysator PY2020 (Shimadzu/Frontier Lab), and by pyrolyzing the dried residue (550 °C) under helium and the pyrolysis products (see Supplementary Table 1).

The wastewater was characterized through thermodesorption gas chromatography mass spectrometry (GC-MS) using Trace GC Ultra / ISQ (Thermo Scientific). The residue was thermally extracted in the thermodesorption unit under helium flow. The components released at 220 °C (10 min holding time) were detected by GC-MS (see Supplementary Table 2). As the wastewater contains a significant number of particles in the microscale range, which are prone to sedimentation, a centrifugation before the separation process was performed, using a Hettich Rotofix 32 A, 220 V centrifuge for 10 min and at 4,000 x $g$. The concentration in the supernatant was 0.03 wt.%.

Differential scanning calorimetry (DSC) analyses were conducted using a NETZSCH DSC 214 instrument. For each analysis, 10 mg of the PMMA sample was precisely measured and loaded into an Al concave pan with a perforated lid. Dynamic measurements were carried out under a $N_2$ atmosphere within a temperature range of 20 °C to 200 °C, employing a heating rate of 10 K min$^{-1}$.

Attenuated total reflectance infrared (ATR-IR) spectroscopy analyses were performed using a Jasco FT/IR-4100 instrument equipped with a PIKE GladiATR ATR-cell featuring a Diamond-Prism. Spectra were obtained in transmission mode with 16 scans and a resolution of 4.0 cm$^{-1}$ within the spectral range of 4500 to 450 cm$^{-1}$.

X-rays Photo electron Spectroscopy (XPS) measurements were taken using a monochromatic X-ray spot with a diameter of 650 μm. All details about the measurement parameters are mentioned in Supplementary Table 7. Prior to the measurements, the samples were pumped out at room temperature to the required 10$^{-8}$ mbar.

### Analysis of the removal efficiency

The removal efficiency of nanoplastic particles was calculated from the weight difference of the dispersion before and after the ePhoam process. 1 mL of the initial dispersion and a range from 1 to 6 mL, depending on the treatment efficiency, of the purified dispersion in the electrolytic cell after particle separation were taken carefully. Then, the solid content was gravimetrically determined, using an analytical balance (Sartorius Quintix), after evaporating the continuous water phase in vacuum for 16 h at 45 °C and 35 mbar (Heraeus Thermo Scientific VT 6060 MBL). The Formula to calculate the removal efficiency and the absolute error is given in Eq. 1.

$$\text{Removal} \pm e_{\text{Removal}} = \left( \frac{w_0 \pm e_{w_0} - w_t \pm e_{w_t}}{w_0 \pm e_{w_0}} \right) \quad (1)$$

Where "Removal $\pm e_{\text{Removal}}$" represents the removal efficiency along with its absolute error; $w_0 \pm e_{w0}$ corresponds to the initial mass fraction of the sample (mg mL$^{-1}$) with its absolute error and $w_t \pm e_t$ denotes the mass fraction of the sample at a specific time $t$ (mg mL$^{-1}$) with its absolute error. The uncertainty arises from limitations in the balance, with the absolute error being 0.1 mg.

Similarly, the removal efficiencies through film and foam were determined. The used electrode was weighed both before and after the process to determine the film mass, while stable foam was gathered on a pre-weighed aluminum plate, with removal efficiency calculated based on the weight difference. Each experiment was conducted at least three times, and the figures present the averages and standard deviations obtained from these repetitions.

## Electrophoretic deposition and particle-stabilized foam process

The electrochemical cell was made of polycarbonate with dimensions of $5 \, cm \times 2 \, cm \times 9 \, cm$ and an operating volume of 80 mL. All experiments were made in batch mode. Stainless steel and mixed metal oxide (MMO) electrodes were tested. MMO electrodes, also known as dimensionally stable anodes (DSA), were based on titanium and coated with iridium mixed oxides. The flat and mesh DSA electrodes had a geometric surface area of $55 \, cm^2$ and $133.5 \, cm^2$ respectively. In the reactor, the anode and cathode were fixed vertically with 1 cm space and were connected to the positive and negative outputs of a DC power supply (MCA series 750–1000, FuG), respectively. The tests were carried out at room temperature. Unless otherwise mentioned, all experiments of colloidal model particles were carried out at room temperature with an initial concentration of 1 wt.%. The colloidal dispersions were ultrasonicated at 1200 W for 5 min prior experiments. A total volume of 20 mL was used for all the model colloidal particles and environmental wastewaters. PMMA model particles were used as a test system to study the mechanism and the impact of different operational parameters. Various concentrations (0.01 to 1 wt.%), electrodes distance (1 to 4 cm), applied voltage (5 to 50 V), separation time (5 to 25 min) were studied to investigate the effect on the efficiency of removal. For the optimal removal in the case of PMMA system, 50 V was applied for 16 min (4 min per cycle), with a concentration of 1 wt.% and an electrode distance of 1 cm for the following model particles too. The conductivity was ≈ $1600 \, \mu S \, cm^{-1}$.

As the different model colloidal particles have different conductivities, pH and electrophoretic mobilities, the applied voltage was adjusted to optimize the efficiency of removal and conduct the experiments with a comparable time per cycle (4 min per cycle). PS-carboxylate, PS-sulfonate and PS- amidine had conductivities of $560 \, \mu S \, cm^{-1}$, $400 \, \mu S \, cm^{-1}$ and $130 \, \mu S \, cm^{-1}$ respectively, which required applying a voltage of 70 V instead of 50 V during a total average time of 20 min (5 cycles for PS- carboxylate and PS-amidine; and 6 cycles for PS-sulfonate). However, in the case of PBMA, the conductivity was 3 mS $cm^{-1}$; thus, the applied voltage was decreased to 32 V during a total time of 20 min too (5 cycles). All experiments have been repeated at least three times and the data presented in the figures is the average with the standard deviation.

## Local pH shift measurement

For the measurement of local pH in the vicinity of both electrodes, the following procedure from[84] was adapted. First, we measured the pH upon water electrolysis using pure distilled water. The pH of the water was adjusted to 2.7 using nitric acid 0.1 M and ionic strength was kept constant by maintaining a conductivity of ≈ $1.65 \, mS \, cm^{-1}$ adjusted using 1 M $CaSO_4$ solution to simulate the properties of a 1 wt.% PMMA dispersion. About 50 μL water sample was taken carefully with a micropipette with sharp tip as close as possible to the surface of both electrodes at different time interval ranging from 30 s to 10 min and transferred onto an ion sensitive field effect transistor pH meter (Model pH BOY® P2, Shendengen Electric Manufacturing Co., Ltd., Tokyo, Japan). At least three measurements were made for each data point.

## Study of bubbles formation on the anode

The bubbles formation was investigated in a transparent cell with 50 mL of tap water. The cathode used was stainless steel, while the anode varied between the three electrodes under consideration. A voltage of 30 V was applied across the electrodes. A high-speed camera acquired the bubble evolution (Motion Blitz EoSens Mini 1, Mikrotron) with a microscopic objective of 50x (L-Plan SLWD 50×/0.45, Nikon). The acquisition rate was 250 fps with a resolution of 756 × 720 pixels. After the recording, the videos were analyzed using the ImageJ software. The average number of bubbles per $mm^2$ and the average detachment diameter were obtained from the videos. The average number of bubbles per $mm^2$ was obtained by counting the bubbles in an $mm^2$ area across 15 s. The average detachment diameter was measured across 20 bubbles.

## Real industrial wastewater system

The industrial wastewater was produced at one cubic meter per hour in a process plant, containing PMMA colloidal particles and surfactant stabilizers in an aqueous continuous phase. Due to the process fluctuations, the solids content of the dispersed components varies between 5 wt.% and 7 wt.%. For the measurements, this industrial colloidal dispersion was diluted to 1 wt.% to facilitate comparison with the above studied model system. The dispersion had a pH value between 5.8 and 6.2, a conductivity between 1000 to $1050 \, \mu S \, cm^{-1}$, an average density between 950 and $1000 \, kg \, m^{-3}$, an average particle size of 130 nm and an average Zeta potential of −49 mV.

The eyeglass and contact lenses polishing wastewater was kindly provided by an optic shop in Blaufelden in Germany. This wastewater was filtered on site using a nylon membrane with a cut-off around 80 μm. It had a supernatant of 0.03 wt.%. The water-based paint was purchased from Marabu Decorlack. The concentration of the paint was determined gravimetrically to 30 wt.% and was diluted to 1 wt.% with ultrapure water for comparison with the other systems. The water-based paint has a zeta potential and particle size of −59.5 mV and 238 nm, respectively. The producer of the water-based paint did not provide the exact ingredients of the product except for two microbial growth controlling compounds (2-methyl-2H-isothiazol-3-one and 1,2-benzisothiazol-3(2H)-one).

The wastewater from water-based paint and polishing of eyeglasses and contact lenses had conductivities of $330 \, \mu S \, cm^{-1}$ and $375 \, \mu S \, cm^{-1}$, respectively. The voltage applied to both systems was 50 V and 70 V, respectively. 5 cycles were needed for eyeglass polishing wastewater compared to 4 cycles for water-based paint, due to the lower concentration of the former (0.03 wt.%) compared to 1 wt.%.

## COD and TOC measurement

Wastewater samples before and after treatment were mixed with potassium dichromate reagent and added to the Thermoreactor RD125 for digestion (150 °C; 2 h) for COD measurements and (120 °C; 1 h) for TOC measurements. After cooling down, the samples were measured in a photometer AL100 to measure the consumption of dichromate during the oxidation process and thus read the COD and TOC values at 400 nm. The measurements were repeated 3 times for each parameter and each sample.

## Continuous pilot plant

The continuous ePhoam setup consists of a rotating DSA anode (Metakem GmbH) with a diameter of 25 cm and a length of 30 cm. The available area is around $2350 \, cm^2$. The cathode is a stainless steel semicircular cut tube with similar curvature to the anode and held by two plastic plates. It has an area around $1480 \, cm^2$. The base forms a trough, where the water can be deposited. The slips in the center of the plastic plates support the anode roller. A motor unit (TXF/005) rotates the roller. The motor was adjusted to have a 0.75 rpm speed. At the bottom of the plastic plates, hose connections are placed to connect a pump. The pump feeds and discharges the polymer dispersion in the system. The anode and cathode are connected to the DC power supply (MCA series 750–1000, FuG). 2.5 L of the desired dispersion was treated in the roller setup.

## Data availability

The data that support the findings of this study are available via Zenodo[85] and from the corresponding author upon request.

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

## Acknowledgements

This project has received funding from the European Union's Horizon 2020 research and innovation programme under the Marie Skłodowska-Curie grant agreement No860720 and by the Deutsche Forschungsgemeinschaft (DFG, German Research Foundation)—Project-ID 416229255—SFB 1411. Herbert Canziani is acknowledged for his help with the synthesis of the PMMA model particle dispersions. Teresa Walter is acknowledged for her help with the visualization of the bubble formation process.

## Author contributions

P.S., N.V. and Y.E.T. initiated the project and supervised the work. A.A., B.L.R. and M.L conducted the experiments. B.L.R performed the experiment of bubbles formation and the removal efficiency from the pilot plant. P.S and Y.E.T. designed the roller setup. A.A. and M.L. performed the experiments of removal efficiencies from the different model systems, and industrial wastewaters. P.S., A.A. and M.L. conducted the cost efficiency and carbon footprint estimation for the process. N.B. and G.B. synthesized and characterized the PS and PBMA model systems. N.B. performed the DSC and FTIR characterization. The manuscript was written by A.A. and N.V. with input from all authors. All authors have given their approval to the final form.

## Funding

## Competing interests

The patent EP 3909918B1 ("Method and Device for the Electrostatic Separation of Polymer Particles from Waste Water") has been granted based on parts of the presented technology: [roller setup for removal of polymer particles] and represented by Evonik Industries AG Patent Association (P.S., Y.E.T., N.V.). Apart from that, the authors declare no competing interests.
