## [Peer Review File · Nature Communications]

Efficient removal of nanoplastics from industrial wastewater through synergetic electrophoretic deposition and particle stabilized foam formationREVIEWER COMMENTS

Reviewer #1 (Remarks to the Author):

This paper reported a method for removing nano-plastics in industrial waste water via electrophoretic deposition, the method is proved to be efficient. This paper is well organized in structure, data is solid and well presented. The problem with this paper is generic: the work is lack of novelty and scientific implication. Electrophoretic deposition (EDP) is a well established technique, and there is no new discovery, new mechanism or new concept in this manuscript. Primarily for this reason, I would not recommend this manuscript to be published in Nature Communication.

Specific comment:

1. As a wastewater nanoplastic removal technique, the paper mentioned the treatment efficiency, but does not give the estimation of energy consumption per unit volume, which is probably the most important factor to be considered in wastewater treatment.
2. Paper describes the electrode used in this work, but the deposition of nanoplastic (as well as other carbon) should influence the area of the electrode, if this will affect the treatment efficiency?
3. To evaluate the method, author should compare it with other nano plastic processing methods (such as electric adsorption, Water Research 2020:1116100).

Reviewer #2 (Remarks to the Author):

Line 47: this statement is valid for free (i.e., non-aggregated) NPs. However, current literature says many of them are likely to be coated with (natural) molecules and other particles. Consequently, this statement must be rephrased, in my opinion. If attached to other materials, NPs could be filtered or even settled in specific environmental compartment and/or in dedicated settling tanks during water treatment – more especially if aggregation is promoted by coagulants/flocculants. In fact, larger plastics such as MPs, after aggregation, are less removed than NPs because larger particles tend to be less incorporated/embedded into an existing floc structure when they are too big. Moreover, the literature shows that environmentally relevant NPs (weathered NPs, not pristine) are more easily aggregated in some systems.

Lines 55-57: again, force fields on small particles can be used – and are efficient - if the colloids are still colloids i.e., not aggregated into larger flocs.

Lines 63-64: EPD works when particles are charged. However, negatively charged colloids (as NPs) could be partially neutralized/stabilized by surrounding molecules. Not sure EPD will work so well in such context.

Line 74: PMMA is not the most common plastics in wastewater, except for industries manufacturing PMMA. Why not test this process (EPD and electrochemical gas bubble) on other common NPs e.g., PS, PE, etc. The annual production of these later plastics are in fact higher than PMMA.

Line 79: space between 1 and μ

Lines 85-87: again, other techniques than EPD could be used here to remove colloidal PMMA: aggregation+settling, aggregation+granular filtration, membrane filtration, degradation via oxidation, etc.

Lines 93-98: negatively charged NPs, via -COOH, are tested here. However, carboxylate sites on NPs could be severally blocked if other molecules/colloids are involved, which is usually the case in the vast majority of industrial and municipal wastewater. Like mentioned above, the EPD process would be affected. Secondly, when plastic surfaces are weathered, they tend to be less hydrophobic; I believe the removal mechanism via oxygen bubble would also be affected in that case.

Figures 1-3 are very elegant and clear.

Figure 2: to reach removals around 80% - which can be accomplished by typical enhanced coagulation - the contact time must be > 25 min. In terms of process capacity, it could be an important issue as it limits the flow rate for a given contactor volume. As mentioned above, I doubt that such process could deal with a very high flow rate at reasonable cost, even after considerable optimization.

Figures 3-4: The authors show that processing the water for 4 cycles increased the recovery. We don't usually have the chance to do this in water treatment, where the chosen technology is a very difficult compromise between cost and performance (recovery in that case). I don't see the point of doing this demonstration via 4 cycles.

Conclusion: If O₂-bubble is a significant mechanism to transport NPs, hence this could be accomplished by simple O₂ injection (or air sparging). In fact, existing literature already reported such mechanism during flotation.

The format of the paper is of high quality and it is very well written. However, the scope of such process is limited to industrial wastewater with a high concentration of colloidal PMMA. This process could not be upscaled to specifically remove NPs in other industrial influent, nor municipal wastewater. And the process capacity could eventually be limited by energy in the system i.e., water treatment plants with more typical NPs concentration or with higher flow rate could not benefit from the proposed technology. Consequently, the industrial application is rather limited.

I'm not sure this paper is suitable for Nat. Comm. or any other very high impact journals. I think the manuscript should be rather submitted into a more technical and water-related journal e.g., Water Research, ES&T, ES&T-Water, or Chemical Engineering Journal.

Reviewer #3 (Remarks to the Author):

This study investigates the removal of nanoplastics by electrochemistry. The removal efficiency is good and the paper is well arranged. However, the novelty and significance of this study are not high and several critical issues need consideration.

1. The so-called "electrostatic separation" is a combination of electroadsorption and electroflotation, which has been well demonstrated in previously studies (Water Research, 2020, 184, 116100.; Clean - Soil, Air, Water 2020, 49, 2000146.; Separation and Purification Technology, 2021, 276, 118877.; Separation and Purification Technology, 2022, 290, 120905.; Chemical Engineering Journal, 2022, 428, 131161.; ACS Omega, 2018, 3, 3, 3357-3364.). In this context, the novelty of this study is not sufficient. In addition, the electrochemical process adopted in this study involves the electrolysis of water, which should be a redox process, and thus the "electrostatic separation" is inappropriate.
2. As demonstrated, both H₂ and O₂ was generated and the amount of H₂ should be twice of O₂, why nanoplastics only attached onto the O₂ bubbles? As the electrode distance was set at 1 cm, it is difficult to identify O₂ and H₂ separately.
3. For the "particle-stabilized foam", the stability of the formed foam should be quantitatively measured.
4. Authors declared that the size and number of bubbles was dependent on electrodes; but previous studies have suggested that the current density plays an important role in bubble formation, which was ignored in this study.
5. In Figure 2b, under 20 V all nanoplastics were recovered by film. Although in this process no bubbles were formed, the water might undergo electrochemical activation and generate radicals that influence the oxidation of nanoplastics and thereby the removal process and mechanism. Thus, more work is needed to identify this process.
6. The effect of electrochemical process on the evolution of nanoplastics' surface chemistry or

degradation should be considered.

7. Line 196-198, the role of leached Fe³⁺ should be discussed, which has shown a typical role in the removal of nanoplastics.

8. In Figure 3c-d, the size of the recovered foam seems has a size of ~0.5 μm, which is larger than the description of 361 nm in the fourth line in the "Results and discussion" section.

9. The applicability of this process for different nanoplastics (size, species) was not investigated.

10. Compared with electrocoagulation/electroflotation process which operates at relatively low potentials, the applied potential in this study is quite high. In this context, is this process economically viable?

Reviewer #4 (Remarks to the Author):

This study developed a novel technology for the removal of nanoplastics from wastewater which combined the electrophoretic deposition processes and the formation of stable, particle-stabilized foams. However, the feasibility and adaptability of this technology on nanoplastic removal from wastewaters were not systematically and sufficiently studied. The in-depth explanations of results and experimental phenomena were not made as well.

1. The authors are suggested to carefully modify the manuscript since a number of typos and grammatical mistakes are found throughout the text. For instance:

1) Figure 2a shows that for a fixed applied voltage, the recovery percentage linearly increases with time 158 but eventually plateaus.

2) Ln 302: 2.3. characterization of PMMA colloidal particles

3) Ln 321: 2.5. Electrostatic separation process:

4) Ln 335: "... adjusted to 2.7 using Nitric acid 0.1M ..."

5)

2. The feasibility of electrophoretic deposition-bubble collection technology needs to be validated. For the removal of nanoplastics from waters, the impacts of size (from tens to hundreds of nanometers), shapes (fiber, sphere, rod, irregularly-shaped debris) and surface-hydrophobicity can significantly change the capture and separation behaviors. Typically, the aggregation and dispersity of PMMA nanoplastics in water or wastewater are still unknown before treatment.

3. For the nanoplastics-polluted wastewater, the interaction of organic matter molecules and nanoplastics influence the performance of treatment efficiency. However, it was not studied in this study.

4. The formation of nanoplastic foams by oxygen microbubbles can be impacted by water chemistry. For instance, the co-existing organic matter and particulate pollutants as well as the change of ionic strength should be investigated for their impact on nanoplastic-foams and thus the efficiency of nanoplastic removal.

5. The cost-efficiency analysis should be made and compared with traditional electrochemical and flotation treatment such that the advantages of the technology proposed in this work can be clarified.

Reply to reviewer comments

Overview and general aspects

In the revised version, we have highlighted changes in yellow and provide a point-by-point response to all reviewer comments below. In particular, in the revised version, we have included the following general points of criticism raised by the reviewers:

- A broad investigation on the universality of our methodology, implemented by successful removal of nanoplastics from a range of different dispersions, by a systematic investigation of model systems, including a second acrylate (polybutylmethacrylate), polystyrene, and different surface chemistries.
- Demonstration of industrial relevance by showing removal of nanoplastics from dispersion paints and wastewater from the eyeglass polishing industry, besides the initial study on industrial wastewater from the PMMA production. All these processes are widely used in industry and or daily life and are known sources for nanoplastic pollution.
- A proof-of-concept on the scalability and practical relevance by demonstrating the transfer of the methodology from a lab-scale batch process to a roller-based system operating in a continuous process via a cylindrical anode.
- More detailed insights and discussion into the scientific aspects underpinning the coupling of electrophoretic deposition, bubble formation, and trapping of particles at the interface of these bubbles.
- A discussion on cost-efficiency and differentiation to existing processes.

Reviewer #1

This paper reported a method for removing nano-plastics in industrial wastewater via electrophoretic deposition, the method is proved to be efficient. This paper is well organized in structure, data is solid and well presented. The problem with this paper is generic: the work is lack of novelty and scientific implication. Electrophoretic deposition (EDP) is a well-established technique, and there is no new discovery, new mechanism or new concept in this manuscript. Primarily for this reason, I would not recommend this manuscript to be published in Nature Communication.

We thank the reviewer for appreciating the organization, presentation, and solid nature of our data. However, we politely disagree with the assessment about the novelty and scientific implication. Based on this criticism, we clearly failed to convey these points in the initial manuscript and have revised the manuscript, accordingly, as detailed below.

In particular, we fully agree that both electrophoretic deposition (EPD) and particle stabilized foam are well-established processes. The novelty of our manuscript is that we apply the techniques in a completely different context. In fact, by focusing on the fate of the dispersion, and not the coating, we provide an entirely new context and open these processes to a very different field of application. In addition, we underline that conventionally, these two processes are investigated and used by very different communities. In fact, the formation of bubbles is one of the key points that decrease the quality of the EPD processes, and much research has been focused on suppressing this bubble formation by water splitting. In our manuscript, in contrast, we take advantage of this typical drawback and demonstrate that it can in fact be a great asset, as it allows coupling the EPD process to particle-stabilized foams, which synergistically combine in a new methodology to remove nanoparticles. Importantly, this synergy is very relevant for the combined processes, since in “normal” dispersion, one cannot apply bubbles to create particle-stabilized foams (see Figure 1). We carefully elaborate on the scientific aspects and provide a complete mechanistic insight on the synergy between the two processes. These rely on local changes in pH triggered by the water splitting reaction, which, in turn, alter the surface chemistry of the particles in a way that they become susceptible to be trapped at the interface of the emergent bubbles. In our view, this combination of a novel process technology that provides a possible solution to the unmet challenge of separating – and recycling – nanoplastics, and a clear mechanistic description underpinning this separation process is novel, innovative, and relevant.

We have significantly amended the introduction of the manuscript to underline these points and describe the innovative aspects more clearly:

P.3-4: A key consideration in typical water-based EPD processes is the prevention of water electrolysis³². The evolution of oxygen and hydrogen bubbles hinders the formation of dense and uniform films,³² and are therefore detrimental for the coating process.

A second fundamental property of colloidal particles is their ability to adsorb to liquid interfaces. This adsorption reduces the air- and oil/water interphase and can therefore lower the total free energy³³. The attachment strength is a strong function of the contact angle³⁴, making it inherently dependent on the surface properties of a particle dispersion^{35–39}. The spontaneous adsorption of colloidal particles to interfaces has long been used to form highly stable emulsions, termed Pickering emulsions^{40,41}, and foams, known as particle-stabilized foams^{42,43}. To efficiently create particle stabilized foams, the surface chemistry of colloidal particles needs to be carefully adjusted in line with the contact-angle dependent attachment strength, typically by controlling the pH^{44–47} or by introducing tailored functional groups to the particle surface⁴³.

In our approach to nanoplastic separation from industrial wastewater, we synergistically combine these two distinct properties of particles in the colloidal range, i.e., their ability to migrate in electric fields, and their propensity to form particle-stabilized foams. We take advantage of the surface charges of nanoplastic particles, which are commonly present due to the addition of ionic stabilizers⁴⁸, charged comonomers⁴⁸, or proteins in biological media¹⁴, and apply an electric field to deposit particles onto an electrode. Contrary to a conventional EPD process, we actively employ the electrochemical gas bubble formation as an additional mechanism for the efficient nanoplastic removal. Importantly, the water electrolysis causes changes in the local pH in the vicinities of the electrodes, affecting the surface properties of the nanoplastic particles close to the provided gas/water interface in the form of the nascent bubbles. In particular, protonation of surface charges renders particles more hydrophobic, making them susceptible to the interfacial adsorption process and therefore enable the formation of a particle-stabilized foam. Thus, this process synergistically combines a typical limitation of EPD processes with the formation of a particle-stabilized foam, colluding to efficiently remove nanoplastic particles from an aqueous dispersion.

Specific comment:

1. As a wastewater nanoplastic removal technique, the paper mentioned the treatment efficiency, but does not give the estimation of energy consumption per unit volume, which is probably the most important factor to be considered in wastewater treatment.

We thank reviewer 1 for bringing up this important point, which clearly is relevant for a practical applicability of the process. We have performed a comprehensive analysis of the energy consumption per unit volume and compared the associated costs and carbon footprints with the conventional treatment of industrial wastewater from PMMA production (incineration). These estimations indicate that the process is economically viable. The relevant section of the main text now reads:

P12: Compared to other electrochemical-driven technologies (Supplementary Table 3) and incineration, which is the current technological treatment of industrial wastewater from emulsion polymerization-based processes, the ePhoam process is economically viable and cheaper, in particular when factoring in the ability to recycle the polymer. Additionally, it combines attractive properties of other electrochemical processes (details in Supplementary Information). Implementing the electrophoretic deposition/ particle stabilized foam formation process would lower the carbon footprint by approximately 77 % compared to the state-of-the-art waste treatment (details in Supplementary Information).

Two complete chapters on the energy estimation and cost of the process, its comparison to the currently applied technology for wastewater treatment, and the results on the carbon footprint are included in the Supplementary Information (Chapter 1.3 and 1.4).

2. Paper describes the electrode used in this work, but the deposition of nanoplastic (as well as other carbon) should influence the area of the electrode, if this will affect the treatment efficiency?

Reviewer 1 is correct that the electrode area (and performance) is affected by the deposited particles. These changes on the electrode surface is in fact among the reasons why we worked with multi-cycles recovery in the batch process at small scale. After a certain operational time (which depends on several parameters such as the electrophoretic mobility of the particles, the applied potential, and the

distance between the two electrodes etc.), the particle deposit on the target electrode becomes sufficiently thick to form an insulating layer. Consequently, the current decreases and the recovery becomes stagnant. This effect is well visible in the data of Figure 2, especially at large voltages. In the case of small scale, we used a vertical parallel flat electrode, and simply reverted to a multi-step process to recover the electrode and continue the process, until the dispersion was sufficiently purified by nanoplastic removal. We have added a discussion for this point at the relevant section of the manuscript:

P.7: Figure 2a shows that for a fixed applied voltage, the recovery percentage linearly increased with time but eventually plateaued. This effect is expected as in EPD with constant voltage, the potential difference between the electrodes remains constant. However, the electric field influencing electrophoresis decreased with deposition time due to the formation of a dense, insulating particle film on the electrode surface.³⁰

P. 8: This cyclic operation also ensured an efficient electrode performance, as particles deposited as a thin film on the electrode surface could be removed to avoid the formation of a passivating layer which reduces the electric field and thus the removal efficiency.

In the revised version, we have further included our approach for process scale-up, which we implement by means of a continuous process with a rotating cylinder electrode. This cylinder is connected to a doctor blade, which collects the deposited particles (and provides them for recycling) and continuously recovers the pristine electrode surface. The main text has been amended as follows:

P.15: While the multi-step processes using the transparent electrochemical cell shown in Figure 1 enabled parameter studies and the fundamental elucidation of the mechanisms underpinning the ePhoam process, it lacks the scalability and throughput required for realistic treatment of industrial wastewater. In a proof-of-principle demonstration, we transferred the concept to a scaled pilot reactor capable of a continuous operation. In this “roller” setup, the anode is implemented as a steel cylinder that continuously rotates against a semicircular, static cathode (Figure 7 a-c). The wastewater can be inserted on one side and removed on the opposite side of the reactor. In the course of operation, the anode rotates, picks up particle deposits, both as foam and thin film, which is subsequently removed from the electrode by a doctor blade. As a result, the removed particles can be recycled, while the electrode surface is recovered and re-immersed. Figure 7 d-f shows photographs of the setup during operation. In this pilot plant, 2.5 L total dispersion can be treated at any time. The recovery efficiency of nanoplastic particles depends on the initial concentration, similar to the model system (Figure S5), and reaches values above 90% for concentrations of 0.5 wt.-% and above, using a 2-hour process with 50 V as an applied voltage and 4.25 mA/cm² as a current density (Figure 7g).

Figure 7: Continuous operation and scale up, implemented by a “roller” pilot plant based on a rotating anode. a-c) Schematic illustration of the separation process. A rotating anode attracts the nanoparticles from an aqueous dispersion (a). The deposits are subsequently transported out of the water phase (b), and collected by a doctor blade (c). The replenished electrode re-immersed into the dispersion for a continuous process. d-f) 3D illustration (d) and photographs of the plant during operation, showing the doctor blade (e) and the removal of particle-stabilized foam (f). g) Impact of initial concentration on the recovery percentage using the roller system.

3. To evaluate the method, author should compare it with other nano plastic processing methods (such as electric adsorption, Water Research 2020:1116100).

We have carefully evaluated the available electrochemical processes used for the treatment of wastewater containing micro and nanoplastics. We have summarized each process in terms of principle, type of polymers treated, advantages and disadvantages as well as the cost analysis. The comparison table is presented in the Supplementary Information, Table S3. In brief, while other techniques exist that can have advantages in some aspects, to this point none combines cost efficiency, broad applicability, continuous operation, and full recyclability of the removed particles without degradation of the polymeric material. These properties make our process well suitable for the industrial application, where wastewater with comparably high polymer content needs to be treated.

Reviewer #2 (Remarks to the Author):

Line 47: this statement is valid for free (i.e., non-aggregated) NPs. However, current literature says many of them are likely to be coated with (natural) molecules and other particles. Consequently, this statement must be rephrased, in my opinion. If attached to other materials, NPs could be filtered or even settled in specific environmental compartment and/or in dedicated settling tanks during water treatment – more especially if aggregation is promoted by coagulants/flocculants. In fact, larger plastics such as MPs, after aggregation, are less removed than NPs because larger particles tend to be less incorporated/embedded into an existing floc structure when they are too big. Moreover, the literature shows that environmentally relevant NPs (weathered NPs, not pristine) are more easily aggregated in some systems.

Lines 55-57: again, force fields on small particles can be used – and are efficient - if the colloids are still colloids i.e., not aggregated into larger flocs.

Reviewer 2 raises an important, yet complex point. Indeed, it is correct that colloidal particles in the presence of biological media rapidly obtain a protein corona, which is likely to occur in any natural system. The nature of this protein corona is complex, highly dependent on the environmental conditions, and itself subject to intense research. In our manuscript, we therefore specifically address industrial wastewater, where the particles are in a more controlled environment with less exposure to biological materials and occur in larger quantities, which makes recyclability a much more important consideration. As we show in the revised version, our process is economically viable, reduces the carbon footprint compared to state-of-the-art wastewater treatment (see new additions to SI), and recovers high quality polymeric materials without degradation (see below and Supplementary Figure 11). Together, we believe that these criteria already constitute an attractive strategy for wastewater treatment. However, the role of the protein corona is of course very relevant and understanding of the process for particle dispersions in biological media would constitute an important next step for a

potential applicability in environmental wastewater, or even for the detection and analysis on nanoplastics. The complexity of these considerations merits a separate, detailed study.

With respect to the agglomeration of colloidal particles with a protein corona, we would like to emphasize that this would render the nanoplastic problem somewhat more tractable since the agglomerates (if sufficiently stable and large) can eventually be captured by conventional microplastic separation processes. That said, several publications report on the presence of colloidal particles in water systems and studied its negative impact on different organisms (Li et al, Separation and identification of nanoplastics in tap water. Environmental research 204, 112134, 2022; Oliveri Conti et al. Micro- and nano-plastics in edible fruit and vegetables. The first diet risks assessment for the general population. Environmental research 187, 109677, 2020). We have added a sentence in the introduction to clarify and distinguish these two points:

P. 2: While some nanoplastics may agglomerate upon exposure to biological media due to the formation of a protein corona, thus allowing the separation with conventional microplastic methodologies^{14 16}, nanoplastic particles have nonetheless been found e.g. in tap water¹⁸ or edible fruits and vegetables¹⁹.

In the light of the comment of reviewer 2, we have rewritten the conclusion section to avoid misunderstandings or a false impression on scope and possibilities of the present manuscript.

P. 16-17: The combination of cost and resource efficient treatment, continuous operation and full recyclability without degradation of the polymer material are important criteria for an industrial integration. Since the separation process is not specific to a certain material, it can be translated to wastewater from different industrial processes, from industrial polishing and cleaning solutions, to dispersion-based paints and coatings. Further perspective applications with a large environmental impact include the treatment of wastewater from the natural latex industry.⁷⁹ A future transfer to nanoplastic detection, analysis, and removal in environmental water can be conceptually envisioned, but necessitates fundamental research and technological developments on understanding the role of protein adsorption and aging of such particle systems in biological environments and on the process efficiency with minute particle concentrations.

Lines 63-64: EPD works when particles are charged. However, negatively charged colloids (as NPs) could be partially neutralized/stabilized by surrounding molecules. Not sure EPD will work so well in such context.

Based on the comments by several reviewers, including reviewer 2 (see below), we have included in the revised version a more systematic study of model systems of different materials and surface properties, as well as several other types of industrial wastewaters (for details, please see reply in the point). These investigations clearly show that our combined EPD/Foam-evolution mechanism can remove particles broadly. Additives and other free molecules present in industrial wastewater do not compromise the process and seemingly only have little impact on the efficiency. As an example, according to the specification of the manufacturer, the dispersion paint for example contains biocides and anti-microbial molecules. Even positively charged colloidal particles can be similarly removed, in this case, using the cathodic reaction. In the most heterogeneous system, an eyeglass and contact lenses wastewater, different ions present in normal tap water, as well as various additives and a range of particles with sizes in the nano- and micrometer range are present. In this case, the recovery is not

as high as in the more homogeneous systems, but this reduction in efficiency can also be caused by the lower concentration and low electrophoretic mobility of the particles.

We have included a discussion on the heterogeneity and the presence of additives in the new section on versatility, as discussed in the following point.

Line 74: PMMA is not the most common plastics in wastewater, except for industries manufacturing PMMA. Why not test this process (EPD and electrochemical gas bubble) on other common NPs e.g., PS, PE, etc. The annual production of these later plastics are in fact higher than PMMA.

We initially chose PMMA as an examples of colloidal particles that are industrially relevant, because it is industrially synthesized in an emulsion polymerization, and thus produces large quantities of nanoparticle-containing wastewater. In contrast, other polymers such as PE are typically polymerized in bulk, solution or via catalytic processes and therefore do not directly create particle-containing wastewater during production.

However, as the reviewer kindly pointed out, there are of course many more plastic systems containing nanoparticles. We have now addressed the remark on the versatility of our process very broadly in the revised manuscript. To this end, we have added systematic investigations on a range of model systems, including different (meth)acrylates to investigate the effect of the glass transition temperature; polystyrene as another common material that is readily available in the form of nanoparticles, and an investigation of the effect of surface chemistry, where we incorporated positively-charged amidine groups.

Furthermore, we complemented these model systems with wastewater from different industrial origin (see below for details and additions). Besides the initial PMMA dispersions, we now demonstrate successful particle separation for dispersion paint and eyeglass polishing wastewater with a very heterogeneous, ill-defined composition. While the details change among the tested systems and process criteria at times need to be adapted (e.g. as the dispersion paints readily form films and do not persist as individual particles due to their low glass transition temperature), we can broadly see an efficient separation in all cases.

The amended section of the main text reads:

P4: We investigate the synergistic electrophoretic deposition and particle stabilized foam formation process (ePhoam) in detail and elaborate on the removal mechanism using a well-defined nanoplastic system functionalized with carboxylic acid groups. Subsequently, we investigate the versatility of the process by changing both material- and surface properties of such model dispersions. Finally, we translate our findings to less defined industrial wastewater, with a primary focus on polymethylmethacrylate (PMMA) nanoplastics.

P 10-11 We subjected a range of different model dispersions to the ePhoam process to probe the versatility of the method, in particular with respect to the material and the surface properties of the dispersions. To this end, we synthesized polystyrene (PS) colloidal particles with different surface functionalities by changing the comonomer in the surfactant-free emulsion polymerization process^{67,68} (Figure 4). Carboxylic acid-functionalized PS nanoplastic dispersions showed a similar behavior compared to the PMMA-based model system, and exhibited recovery efficiencies of 96% after 5 operation cycles, both by anodic deposit and particle-stabilized foam (Figure 4a). Sulfonate-functionalized PS nanoplastic particles, which also have a negative surface charge (Figure S9), showed a comparable recovery efficiency with 92% total recovery after 5 cycles (Figure 4b). In this case, the initial recovery at the first operation cycles was lower, and a higher voltage (70 V) was required. These differences may be caused by the more hydrophilic nature of the sulfonate group that

requires higher proton concentration in the vicinity of the anode to efficiently form a particle-stabilized foam (Figure S9). Positively-charged, amidine-functionalized PS nanoplastic particles also exhibited a pH-dependent change in surface properties, albeit with a decrease towards increasing pH values as the amidine groups become neutralized by deprotonation (Figure S9). In this case, the separation achieved comparable yields but the process was inverted, with the removal occurring at the cathode (70 V) in the form of deposited films and a particle-stabilized foam formed from the evolving hydrogen (Figure 4c). Finally, we probed a third material class by synthesizing poly(butylmethacrylate) (PBMA) colloidal dispersions with carboxylic acid surface functionalities⁶⁹. In this case, the surface properties are similar to the initial PMMA-based dispersion and both materials are methacrylates, but differ in their glass transition temperature (T_g). While PMMA has a T_g of ~ 105 °C, the T_g of PBMA is close to room temperature ($T_g \sim 33$ °C)⁷⁰, making the particles prone to film formation during operation. As a result, upon the separation, this dispersion only formed an unstable foam and the deposit on the anode was rather transparent. Both observations relate to the low T_g , which causes the particles to sinter during the process, leading to a homogeneous film on the anode and a mechanically less stable foam. Consequently, the recovery efficiency was lower and reached 63% after 5 cycles (Figure 4d). Noteworthy, this dispersion had a higher conductivity, and thus required a lower voltage (32 V) for the separation process. These experiments demonstrate that the separation process can be applied to a range of different materials and surface properties, but requires adapting of the process conditions to the specific characteristics (surface charge, T_g , conductivity).

Figure 4: Recovery of model nanoplastic particles with different material and surface properties by the electrophoretic deposition/ particle stabilized foam formation process. a) Polystyrene-carboxylate particles; b) Polystyrene-sulfonate particles; c) Polystyrene-amidine particles, d) Polybutylmethacrylate (PBMA) particles.

Line 79: space between 1 and μ

Corrected. The thorough reading of the manuscript by reviewer 2 is much appreciated.

Lines 85-87: again, other techniques than EPD could be used here to remove colloidal PMMA: aggregation+settling, aggregation+granular filtration, membrane filtration, degradation via oxidation, etc.

The reviewer is correct that there are other strategies to remove nanoplastic particles. We have therefore updated the introduction section to provide a fuller picture and used the SI for a comprehensive overview of the different techniques. However, there are several caveats with all of the mentioned processes. Aggregation-based processes rely on inherently instable colloidal systems (which, technically speaking, would not be nanoplastics anymore), or demands the manipulation of the nanoplastic particles via additives in dispersion, which necessitates very large amounts of chemical additives, which may cause their own separation problems. Membrane filtration can suffer from clogging, and spatially separates nanoparticles, but does not recover them from dispersion (hence, does not allow for recycling). Finally, chemical degradation is more of a brute force approach that involves harsh conditions, and cannot be used for recycling.

P2: Agglomeration can also be induced by manipulation of the dispersions by the means of ionic or reactive additives, e.g. in electrocoagulation²⁰⁻²³ or ozonation processes²⁴, enabling the use of conventional removal techniques established for microplastics²⁵. Membrane filtration is yet another strategy to remove such small particles, but suffers from clogging and does not fully remove the particles from the aqueous phase²⁶

In contrast, we believe that our process provides an interesting combination of attractive properties, including the ability to fully recycle the product. In the revised version, we have highlighted this capacity more clearly. We have investigated the materials properties of the industrial PMMA wastewater and showed that the polymer does not significantly degrade in the process. Furthermore, we show in a cost analysis that this recycling makes the process economically and ecologically viable (see answers above and Supplementary Information, chapter 1.3 and 1.4):

P12: SEM images and photographs of the particles recovered from the model dispersions (Figure 3) already indicated that the particles remain intact and clean after separation and may thus be recycled in the industrial production process. To further confirm this recycling potential, we investigated the polymeric properties of the polymer particles within the industrial wastewater with respect to the molecular weight using gel permeation chromatography (Figure S11, Table S5). These measurements did not show any appreciable degradation of the polymer, and thus indicate the possibility to recycle the recovered material.

Figure S11. Gel permeation chromatography of PMMA particles recovered from industrial wastewater before and after the separation process.

Lines 93-98: negatively charged NPs, via -COOH, are tested here. However, carboxylate sites on NPs could be severely blocked if other molecules/colloids are involved, which is usually the case in the vast majority of industrial and municipal wastewater. Like mentioned above, the EPD process would be affected. Secondly, when plastic surfaces are weathered, they tend to be less hydrophobic; I believe the removal mechanism via oxygen bubble would also be affected in that case.

This is an important point for the applicability of the process, but should be separated into two distinct parts. First, as outlined above, the fate of nanoplastics in the environment is complex, subject to conditions (weathering) and protein corona formation. All these will certainly affect the ability to interfacially adsorb the particles and thus remove them via foam formation. We believe, however, that these factors necessitate a thorough study that is beyond the scope of this manuscript and have added a section in the conclusion to outline the possibility, but also the challenges in this regard:

P17: A future transfer to nanoplastic detection, analysis, and removal in environmental water can be conceptually envisioned, but necessitates fundamental research and technological developments on understanding the role of protein adsorption and aging of such particle systems in biological environments and on the process efficiency with minute particle concentrations.

Here, we aim to generally introduce the process and show its applicability in the processing of industrial wastewater downstream of the production. To this end, we initially demonstrated the use of PMMA from an industrial plant, but have now significantly extended the scope. In particular, we have shown that the recovered material does not degrade (see above), and can thus be recycled in the process. We have further shown the cost, and energy efficiency of our process in an industrial setting (see above), and the ability to scale the process via a continuous operation (see also above, and below). In addition, we have now incorporated a range of model dispersions with different material- and surface properties (see above), as well as industrial wastewaters from different industries, including polishing solutions and dispersion paints. In all cases, the formulations are significantly more complex compared to our well-behaved and controlled model systems, and molecular additives in the form of surfactants (PMMA), biocides and antimicrobial materials (paint), and other additives (eye glass polish) are present in all cases. Despite this complexity, our data shows that we can remove significant amounts of the nanoplastic particles in all cases, underlining the applicability of our technology to real world scenarios. The amended section in the manuscript reads:

P13-14: We investigate the versatility of the technology by subjecting wastewater from different industrial applications and processes to the ePhoam process. Figure 6a-c) shows the removal of nanoplastic particles from a commercial dispersion paint. Dispersion paints are water-based materials that contain polymeric binder particles, typically with a size between 80 nm to 1000 nm⁷¹, and inorganic pigment particles⁷². They are ubiquitously used as indoor wall paints and are applied in volumes of approx. 17 million tons per year⁷³. When disposed inadequately, such paints form a large source of nanoplastic particles that are released into the environment. In the Netherlands, a release of approx. 16 tons of plastic particles is estimated from washing out paint brushes alone⁷⁴. Ultimately, the released particles find their way into aquatic ecosystems and enter the food chain⁷⁵. The challenge in separating such dispersion paint-based nanoplastics is the nature of the binder component itself, which is designed for an efficient film-forming ability⁷¹. This property, typically

associated with a low glass transition temperature, can prevent the formation of particle-stabilized foams, as shown above for the pBMA model colloid. Nevertheless, the separation process was successful in removing approximately 92% of all particles from the dispersion, despite the challenges posed by the multi-component systems, the molecular additives present (including antimicrobial additives, as specified by the manufacturer), and the chemical nature of the binder (see Figure 6c). The decreased particle content can be directly observed in photographs of the dispersion after the separation process, as evidenced by the absence of color and significant reduction in opacity (Figure 6b).

Finally, Figure 6d-f shows the results of the removal of micro- and nanoparticles from the wastewater of industrial eye glass and contact lens polishing, obtained from a company focusing on optical products. During this process, polymeric materials are released from the product to be polished in the form of micro- and nanoparticles with ill-defined shapes^{76,77}. This wastewater thus constitutes the most heterogeneous system to be purified. SEM analysis of this wastewater confirmed this heterogeneity and showed the presence of fragments, fibers, and spherical particles with a broad size distribution and a large population around 300 nm – 70 nm (Supplementary Figure 12). The materials composition was found to be heterogeneous and included polycarbonate (PC), poly (allyldiglycol carbonate (PADC), polyurethane (PU), as well as thiourethane-based additives, as shown in Supplementary Figures 12-14 using Fourier-transform infrared (FTIR) spectroscopy, pyrolysis-gas chromatography mass spectroscopy, and thermodesorption-gas chromatography-mass spectrometry, respectively. Figure 5f shows that even in this least-defined wastewater, approx. 65% of all particles were recovered. We identify the low initial concentration of solids within the wastewater (0.03 wt.-%) as the primary reason for this comparably low recovery. Fewer particles are available to stabilize the bubble interfaces, rendering the formation of particle-stabilized foam less efficient⁷⁸. While these results are in line with recent results for an oil-based floatation process⁷⁷ and our own investigation on the concentration-dependent separation efficiency (Figure S5), they underline the effectiveness of the ePhoam process in achieving significant reduction of nanoplastic content even under the least defined and dilute conditions.

Figure 6: Application of the electrophoretic deposition/ particle stabilized foam formation process to wastewater arising from different industrial products and processes, showing SEM images of the particles, the wastewater dispersions before and after treatment, and the treatment efficiency as a number of treatment cycles. a-c) Wastewater from blue dispersion paint. d-f) Wastewater from industrial glass polishing processes.

Figures 1-3 are very elegant and clear.

Many thanks, appreciated.

Figure 2: to reach removals around 80% - which can be accomplished by typical enhanced coagulation – the contact time must be > 25 min. In terms of process capacity, it could be an important issue as it limits the flow rate for a given contactor volume. As mentioned above, I doubt that such process could deal with a very high flow rate at reasonable cost, even after considerable optimization.

Figures 3-4: The authors show that processing the water for 4 cycles increased the recovery. We don't usually have the chance to do this in water treatment, where the chosen technology is a very difficult compromise between cost and performance (recovery in that case). I don't see the point of doing this demonstration via 4 cycles.

We appreciate this point, which has been identified by different reviewers. Our initial strategy was to demonstrate the general feasibility of the process by means of a model setup, without focusing on scalability, cycles, etc. In the light of the reviewer comments, in the revised version, we have now added a section on scalability and continuous operation (shown as Figure 7), which is described in detail in the answer to reviewer 1. Our strategy in the manuscript is now as follows. We first use defined model systems, both for the dispersions and for the setup to establish the process, focus on the process conditions, and explore the mechanistic details. For this, the transparent cell is very well suitable because it allows direct observation of the process, including the direct visualization of the nascent bubbles by a microscope lens with a large focal length. Next, we now establish the universality of the process in different industrial wastewater settings (see answer above) and then, as a proof of principle, we establish the scalability and continuous operation. At present, we are continuing to further develop this potential and have recently acquired a larger roller setup with the ability to operate with 10 L of dispersion. This will serve as a demonstrator for a planned transfer to a pilot plant to test continuous operation in an industrial setting:

Figure for review: Scalability of the ePhoam process.

Conclusion: If O₂-bubble is a significant mechanism to transport NPs, hence this could be accomplished by simple O₂ injection (or air sparging). In fact, exiting literature already reported such mechanism during flotation.

While this would be the most convenient case, unfortunately, simply injecting oxygen (or even air) does not produce a particle-stabilized foam and therefore cannot be used as a separation mechanism. We investigated this aspect in Figure 1 of the initial manuscript by performing exactly this experiment. For a successful removal of particles in the form of particle-stabilized foams in our setup, two processes need to collude: the electrode reaction locally changes the pH, as protons

are formed at the anode (and, if one separates positively-charged particles, as we now have included, hydroxy ions similarly manipulate the pH in the direct vicinity of the cathode). This local change in pH changes the surface properties of the particles, as surface charges can be removed by protonation/deprotonation, respectively. In turn, this change in surface properties renders the particles more hydrophobic, by this increases their adhesion strength to the generated air/water interface, and thus leads to a particle-stabilized foam. Importantly, and in contrast to many other processes, this process does not require the addition of additional components to manipulate the surface properties, an important requirement for applications on larger scales. While all the data was already included in Figure 1, we apparently failed to discuss these important synergies in sufficient clarity and have now amended the main text accordingly:

P3: A second fundamental property of colloidal particles is their ability to adsorb to liquid interfaces. This adsorption reduces the air- and oil/water interphase and can therefore lower the total free energy³³. The attachment strength is a strong function of the contact angle³⁴, making it inherently dependent on the surface properties of a particle dispersion³⁵⁻³⁹. The spontaneous adsorption of colloidal particles to interfaces has long been used to form highly stable emulsions, termed Pickering emulsions^{40,41}, and foams, known as particle-stabilized foams^{42,43}. To efficiently create particle stabilized foams, the surface chemistry of colloidal particles needs to be carefully adjusted in line with the contact-angle dependent attachment strength, typically by controlling the pH⁴⁴⁻⁴⁷ or by introducing tailored functional groups to the particle surface⁴³. In our approach to nanoplastic separation from industrial wastewater, we synergistically combine these two distinct properties of particles in the colloidal range, i.e., their ability to migrate in electric fields, and their propensity to form particle-stabilized foams. We take advantage of the surface charges of nanoplastic particles, which are commonly present due to the addition of ionic stabilizers⁴⁸, charged comonomers⁴⁸, or proteins in biological media¹⁴, and apply an electric field to deposit particles onto an electrode. Contrary to a conventional EPD process, we actively employ the electrochemical gas bubble formation as an additional mechanism for the efficient nanoplastic removal. Importantly, the water electrolysis causes changes in the local pH in the vicinities of the electrodes, affecting the surface properties of the nanoplastic particles close to the provided gas/water interface in the form of the nascent bubbles. In particular, protonation of surface charges renders particles more hydrophobic, making them susceptible to the interfacial adsorption process and therefore enable the formation of a particle-stabilized foam. Thus, this process synergistically combines a typical limitation of EPD processes with the formation of a particle-stabilized foam, colluding to efficiently remove nanoplastic particles from an aqueous dispersion.

P5: In the absence of an applied voltage (and thus, water splitting reactions and pH-change), gas bubbles can be externally added by bubbling oxygen into the particle dispersion, resembling the strategy of a floatation process. However, no foam formation was observed (Figure 1f), which is attributed to the hydrophilic character of typical colloidal dispersions prepared by emulsion polymerization. They carry surface charges in the form of surfactants or surface-bound charged groups to ensure colloidal stability and therefore do not adsorb strongly to the air/water interface³⁴. Consequently, destabilization of the particles must be ensured for foam formation. Only during water electrolysis, the gas evolution was accompanied by the occurrence of pH changes in the vicinity of the electrodes, which altered the surface properties of the particles in situ without the need of any additives, and increased their adhesion strength to the gas/water interface^{33,34,37,57}. This key difference leads to the formation of a persistent, particle-stabilized foam at the anode

(Figure 1g), which can be easily removed from the dispersion (Figure 1a and Figure S3) and remained intact after drying (Figure S4, Supplementary Movie 1).

The format of the paper is of high quality and it is very well written. However, the scope of such process is limited to industrial wastewater with a high concentration of colloidal PMMA. This process could not be upscaled to specifically remove NPs in other industrial influent, nor municipal wastewater. And the process capacity could eventually be limited by energy in the system i.e., water treatment plants with more typical NPs concentration or with higher flow rate could not benefit from the proposed technology. Consequently, the industrial application is rather limited. I'm not sure this paper is suitable for Nat. Comm. or any other very high impact journals. I think the manuscript should be rather submitted into a more technical and water-related journal e.g., Water Research, ES&T, ES&T-Water, or Chemical Engineering Journal.

We appreciate the positive remarks on our manuscript with respect to quality. However, we politely disagree with the assessment on applicability and impact. In the light of the significant revisions, we believe that we have demonstrated the key points of criticism by demonstrating:

- scaling of the process and integration of a continuous operation;
- demonstration of performance for a range of different model systems
- broad applicability to other industrial wastewaters, including very widely applied industrial products such as paints
- an assessment of energy, cost and environmental impact of our system (all compare well with the established technology of wastewater treatment within plants of our partners)
- Demonstration of recyclability by showing that the material does not degrade in the process

Reviewer #3 (Remarks to the Author):

This study investigates the removal of nanoplastics by electrochemistry. The removal efficiency is good and the paper is well arranged. However, the novelty and significance of this study are not high and several critical issues need consideration.

1. The so-called “electrostatic separation” is a combination of electroadsorption and electroflotation, which has been well demonstrated in previously studies (Water Research, 2020, 184, 116100.; Clean – Soil, Air, Water 2020, 49, 2000146.; Separation and Purification Technology, 2021, 276, 118877.; Separation and Purification Technology, 2022, 290, 120905.; Chemical Engineering Journal, 2022, 428, 131161.; ACS Omega, 2018, 3, 3, 3357–3364.). In this context, the novelty of this study is not sufficient. In addition, the electrochemical process adopted in this study involves the electrolysis of water, which should be a redox process, and thus the “electrostatic separation” is inappropriate.

We thank the reviewer for this remark. In an effort to focus our introduction in the initial manuscript we have clearly shortened the discussion about other approaches, which we have now included. We have also added a comprehensive overview of the different deposition processes and their advantages/disadvantages (see Supplementary Table S3). However, we underline that our process is unique in the combination of deposition and the in-situ generation of particle-stabilized foam, which arises without the requirement of any additional additives (such as metal ions released during electrocoagulation, as in the publications mentioned by the reviewer) by exploiting the local changes in the properties of the dispersion, as discussed in detail in the answer to reviewer 2. The revised manuscript now reads:

P2: While some nanoplastics may agglomerate upon exposure to biological media due to the formation of a protein corona, thus allowing the separation with conventional microplastic methodologies¹⁴⁻¹⁶, nanoplastic particles have nonetheless been found e.g. in tap water¹⁸ or edible fruits and vegetables¹⁹. Agglomeration can also be induced by manipulation of the dispersions by the means of ionic or reactive additives, e.g. in electrocoagulation²⁰⁻²³ or ozonation processes²⁴, enabling the use of conventional removal techniques established for microplastics²⁵. Membrane filtration is yet another strategy to remove such small particles, but suffers from clogging and does not fully remove the particles from the aqueous phase²⁶. A potentially more elegant solution to the nanoplastic problems is to take advantage of their colloidal properties, i.e., to generate force fields in which these small particles can move, and to control their interactions, so that the particles can be efficiently collected.

P13: Compared to other electrochemical-driven technologies (Supplementary Table 3) and incineration, which is the current technological treatment of industrial wastewater from emulsion polymerization-based processes, the ePhoam process is economically viable and cheaper, in particular when factoring in the ability to recycle the polymer. Additionally, it combines attractive properties of other electrochemical processes (details in Supplementary Information). Implementing the electrophoretic deposition/particle stabilized foam formation process would lower the carbon footprint by approximately 77 % compared to the state-of-the-art waste treatment (details in Supplementary Information).

In addition, we thank the reviewer for the comment about the title. Indeed, the original title may have been a bit oversimplistic, as remarked by the reviewer. We therefore suggest to change the title to better reflect the actual process steps:

New title: Efficient removal of nanoplastics from industrial wastewater through synergetic electrophoretic deposition and particle-stabilized foam formation.

2. As demonstrated, both H₂ and O₂ was generated and the amount of H₂ should be twice of O₂, why nanoplastics only attached onto the O₂ bubbles? As the electrode distance was set at 1 cm, it is difficult to identify O₂ and H₂ separately.

Reviewer 3 is correct that hydrogen is generated in twice the molar value compared to oxygen. However, as we detail in our comment to the remark of reviewer 2, gas alone is not the key factor to form particle-stabilized foam and thus remove particles more efficiently from dispersion. The key is the synergetic combination between the electrochemical reactions occurring at the electrodes, and the gas evolution. The electrochemistry very locally changes the pH, which, in turn, affects the surface properties of the particles. Depending on their chemical nature, they are rendered more hydrophobic by protonation in the vicinity of the anode (negatively charged particles), where the oxygen evolution reaction produces these protons; or by deprotonation in the vicinity of the cathode (positively charged particles), where the hydrogen evolution reaction provides the required hydroxy ions. Thus, negatively charged particles deposit at the anode, and form foam using the evolving oxygen, while positively-charged particles deposit at the cathode, and form foam using the evolving hydrogen. While it is not easy to distinguish which gas makes the foam in the cell, the deposits and attached foams clearly shows which electrode dominates the process. We have clarified these properties in the introduction and in the text in the section on the new particle systems:

P3: A second fundamental property of colloidal particles is their ability to adsorb to liquid interfaces. This adsorption reduces the air- and oil/water interphase and can therefore lower the total free energy³³. The attachment strength is a strong function of the contact angle³⁴, making it inherently dependent on the surface properties of a particle dispersion^{35–39}. The spontaneous adsorption of colloidal particles to interfaces has long been used to form highly stable emulsions, termed Pickering emulsions^{40,41}, and foams, known as particle-stabilized foams^{42,43}. To efficiently create particle stabilized foams, the surface chemistry of colloidal particles needs to be carefully adjusted in line with the contact-angle dependent attachment strength, typically by controlling the pH^{44–47} or by introducing tailored functional groups to the particle surface⁴³.

In our approach to nanoplastic separation from industrial wastewater, we synergistically combine these two distinct properties of particles in the colloidal range, i.e., their ability to migrate in electric fields, and their propensity to form particle-stabilized foams. We take advantage of the surface charges of nanoplastic particles, which are commonly present due to the addition of ionic stabilizers⁴⁸, charged comonomers⁴⁸, or proteins in biological media¹⁴, and apply an electric field to deposit particles onto an electrode. Contrary to a conventional EPD process, we actively employ the electrochemical gas bubble formation as an additional mechanism for the efficient nanoplastic removal. Importantly, the water electrolysis causes changes in the local pH in the vicinities of the electrodes, affecting the surface properties of the nanoplastic particles close to the provided gas/water interface in the form of the nascent bubbles. In particular, protonation of surface charges renders particles more hydrophobic, making them susceptible to the interfacial adsorption process and therefore enable the formation of a particle-stabilized foam. Thus, this process synergistically combines a typical limitation of EPD processes with the formation of a particle-stabilized foam, colluding to efficiently remove nanoplastic particles from an aqueous dispersion.

P 10: Positively-charged, amidine-functionalized PS nanoplastic particles also exhibited a pH-dependent change in surface properties, albeit with a decrease towards increasing pH values as the amidine groups become neutralized by deprotonation (Figure S9). In this case, the separation achieved comparable yields but the process was inverted, with the removal occurring at the cathode (70 V) in the form of deposited films and a particle-stabilized foam formed from the evolving hydrogen (Figure 4c).

3. For the “particle-stabilized foam”, the stability of the formed foam should be quantitatively measured.

Particle-stabilized foams are solid materials that are essentially stable unless mechanically destroyed (see, e.g. Gonzenbach et al., *Angew. Chem.* 2006, Ref 42). This property arises from the strong interaction of the particles, first with the air/water interface, and subsequently, with each other upon drying. This is true and we observed foam stable for several month (stored simply in the laboratory, see Figure S4), but this long-term stability is not required for our process. In the separation, it is sufficient that the foam is stable during the experiments. This is several minutes in case of the model electrolytic cell, and even less for the new, scalable continuous setup. This stability was reached for all foams found in our study, with the notable exception of low Tg materials (e.g. dispersion paint and poly(butylmethacrylate) dispersions). In these cases, a particle-stabilized foam cannot form since the low glass transition temperature induces film formation upon drying. The particles are therefore not present in a (hard) particle form, but rather form a very thin, continuous membrane which lacks mechanical stability. In that case, the recovery efficiency can be decreased as the removal pathway via particle-stabilized foam is not possible. We have commented on these points in the revised version:

P. 10: Finally, we probed a third material class by synthesizing poly(butylmethacrylate) (PBMA) colloidal dispersions with carboxylic acid surface functionalities⁶⁹. In this case, the surface properties are similar to the initial PMMA-based dispersion and both materials are methacrylates, but differ in their glass transition temperature (T_g). While PMMA has a T_g of ~ 105 °C, the T_g of PBMA is close to room temperature ($T_g \sim 33$ °C)⁷⁰, making the particles prone to film formation during operation. As a result, upon the separation, this dispersion only formed an unstable foam and the deposit on the anode was rather transparent. Both observations relate to the low T_g , which causes the particles to sinter during the process, leading to a homogeneous film on the anode and a mechanically less stable foam. Consequently, the recovery efficiency was lower and reached 63% after 5 cycles (Figure 4d).

4. Authors declared that the size and number of bubbles was dependent on electrodes; but previous studies have suggested that the current density plays an important role in bubble formation, which was ignored in this study.

We agree that the size and number of bubbles are also dependent on the current densities. In our case, our strategy was to explore the general scope and effectivity of the process. To this end, we first optimized the voltage formed for bubble formation, and then fixed it to investigate electrode materials and geometries. The extraction of bubble size from these parameters, in our view, accurately supports our findings (i.e. why the DSA electrodes outperform conventional stainless steel). Of course, it is our hope that the idea will be picked up by different communities, including electrochemists, polymer scientists, environmental scientists, process engineers etc, to elucidate all process parameters, investigate different electrode materials and other process considerations. These optimizations hold the potential to further refine the process and provide ideal conditions for different pollution scenarios.

5. In Figure 2b, under 20 V all nanoplastics were recovered by film. Although in this process no bobbles were formed, the water might undergo electrochemical activation and generate radicals that influence the oxidation of nanoplastics and thereby the removal process and mechanism. Thus, more work is needed to identify this process.

This is a very relevant point by the reviewer and we have now included an analysis of the PMMA polymers before and after treatment. We find that the polymers do not show any significant sign of decomposition in our analysis.

The amended sections of the manuscript now read:

P12: SEM images and photographs of the particles recovered from the model dispersions (Figure 3) already indicated that the particles remain intact and clean after separation and may thus be recycled in the industrial production process. To further confirm this recycling potential, we investigated the polymeric properties of the polymer particles within the industrial wastewater with respect to the molecular weight using gel permeation chromatography (Figure S11, Table S5). These measurements did not show any appreciable degradation of the polymer, and thus indicate the possibility to recycle the recovered material.
not show any appreciable degradation of the polymer, and thus indicates the possibility to recycle the recovered material.

Figure S11. Gel permeation chromatography of PMMA particles recovered from industrial wastewater before and after the separation process.

Table S5

Sample	Mn (g/mol)	Mw(g/mol)	D
PMMA polymer before separation- Dispersion	24400	65600	2.68
PMMA polymer after separation- recovered Powder	22700	66100	2.91

6. The effect of electrochemical process on the evolution of nanoplastics' surface chemistry or degradation should be considered.

In the revised manuscript, we have addressed this concern in two ways. First, as discussed above, we have evaluated the materials properties of the PMMA polymer and did not find any appreciable signs for degradation. Secondly, we have expanded the discussion on the change of particle hydrophobicity, which, indeed, is triggered by the electrochemical process. This, in turn, provides the basis to efficiently form particle stabilized foam. This synergy between the two process steps apparently was not described by us in sufficient clarity and we have amended the main text as follows to clarify:

P3: A second fundamental property of colloidal particles is their ability to adsorb to liquid interfaces. This adsorption reduces the air- and oil/water interphase and can therefore lower the total free

energy³³. The attachment strength is a strong function of the contact angle³⁴, making it inherently dependent on the surface properties of a particle dispersion^{35–39}. The spontaneous adsorption of colloidal particles to interfaces has long been used to form highly stable emulsions, termed Pickering emulsions^{40,41}, and foams, known as particle-stabilized foams^{42,43}. To efficiently create particle stabilized foams, the surface chemistry of colloidal particles needs to be carefully adjusted in line with the contact-angle dependent attachment strength, typically by controlling the pH^{44–47} or by introducing tailored functional groups to the particle surface⁴³.

In our approach to nanoplastic separation from industrial wastewater, we synergistically combine these two distinct properties of particles in the colloidal range, i.e., their ability to migrate in electric fields, and their propensity to form particle-stabilized foams. We take advantage of the surface charges of nanoplastic particles, which are commonly present due to the addition of ionic stabilizers⁴⁸, charged comonomers⁴⁸, or proteins in biological media¹⁴, and apply an electric field to deposit particles onto an electrode. Contrary to a conventional EPD process, we actively employ the electrochemical gas bubble formation as an additional mechanism for the efficient nanoplastic removal. Importantly, the water electrolysis causes changes in the local pH in the vicinities of the electrodes, affecting the surface properties of the nanoplastic particles close to the provided gas/water interface in the form of the nascent bubbles. In particular, protonation of surface charges renders particles more hydrophobic, making them susceptible to the interfacial adsorption process and therefore enable the formation of a particle-stabilized foam. Thus, this process synergistically combines a typical limitation of EPD processes with the formation of a particle-stabilized foam, colluding to efficiently remove nanoplastic particles from an aqueous dispersion.

7. Line 196-198, the role of leached Fe³⁺ should be discussed, which has shown a typical role in the removal of nanoplastics.

We appreciate this remark. Indeed, our stainless-steel electrodes release Fe³⁺ ions, which are also used in electrocoagulation processes. However, the performance of DSA electrodes in our case was better compared to stainless steel, even though in this case, no ion release was detected. This demonstrates that electrocoagulation is not a dominating effect in our system. We have amended the discussion with respect to this point:

P-8: The superior performance of the DSA electrodes, which did not release any ions, shows that electrocoagulation^{20–23}, a phenomenon based on dispersion destabilization triggered by the release of multivalent ions, is not a dominant mechanism in our process.

8. In Figure 3c-d, the size of the recovered foam seems has a size of ~0.5 μm, which is larger than the description of 361 nm in the fourth line in the “Results and discussion” section.

The size of the dispersion was measured by image analysis of SEM images dried on a flat, solid silicon wafer substrate, and to us therefore seems more reliable. The different angle of the foam-based SEM image may be slightly misleading in judging the size.

9. The applicability of this process for different nanoplastics (size, species) was not investigated.

In the revised version, we have now included a range of different dispersions – ranging from model colloids with different material and surface chemistry (new Figure 4), as well as industrial wastewater from different sources (new Figure 6). This probes a large range of sizes and species.

In particular, we used wastewater from the polishing of eye glass and contact lenses, with a broad range of sizes. We have added a characterization of these less defined materials as Supplementary Information:

Figure S12: Particle size and shape characterization of the wastewater generated in the eye glass and contact lens polishing. a-f) SEM images of the particles found in the wastewater, showing different shapes and structures; g,h) SEM images of differently-sized spherical particles. i) Particle size distribution measured by Partica LA-950V2 laser diffraction particle size distribution analyzer.

10. Compared with electrocoagulation/electroflotation process which operates at relatively low potentials, the applied potential in this study is quite high. In this context, is this process economically viable?

We thank reviewer 3 for bringing up this important point, which clearly is relevant for a practical applicability of the process. We have performed a comprehensive analysis of the energy consumption per unit volume, and compared the associated costs and carbon footprints with the conventional treatment of industrial wastewater from PMMA production (incineration). These estimations indicate that the process is economically viable. The relevant sections of the main text now reads:

P13: Compared to other electrochemical-driven technologies (Supplementary Table 3) and incineration, which is the current technological treatment of industrial wastewater from emulsion polymerization-based processes, the ePhoam process is economically viable and cheaper, in particular when factoring in the ability to recycle the polymer. Additionally, it combines attractive properties of other electrochemical processes (details in Supplementary Information). Implementing the electrophoretic deposition/particle stabilized foam formation process would lower the carbon footprint by approximately 77 % compared to the state-of-the-art waste treatment (details in Supplementary Information).

Two complete chapters on the energy estimation and cost of the process, its comparison to the currently applied technology for wastewater treatment, and the results on the carbon footprint are included in the Supplementary Information (Chapter 1.3 and 1.4).

Reviewer #4 (Remarks to the Author):

This study developed a novel technology for the removal of nanoplastics from wastewater which combined the electrophoretic deposition processes and the formation of stable, particle-stabilized foams. However, the feasibility and adaptability of this technology on nanoplastic removal from wastewaters were not systematically and sufficiently studied. The in-depth explanations of results and experimental phenomena were not made as well.

In the revised version of the manuscript, we have extensively added data and discussion on the mechanistic insights and explanations and performed a much more systematic study by screening through different particle systems, exploring the effect of surface properties, materials, polymer parameters on the separation. In addition, we amended the manuscript with more practical considerations using less-defined industrial wastewater, and a perspective on a scalable process. We believe that these extensive revisions provide sufficient argument for the feasibility, adaptability, and explanations of our technology.

1. The authors are suggested to carefully modify the manuscript since a number of typos and grammatical mistakes are found throughout the text. For instance:

- 1) Figure 2a shows that for a fixed applied voltage, the recovery percentage linearly increases with time 158 but eventually plateaus.
- 2) Ln 302: 2.3. characterization of PMMA colloidal particles
- 3) Ln 321: 2.5. Electrostatic separation process:
- 4) Ln 335: "... adjusted to 2.7 using Nitric acid 0.1M ..."
- 5)

We appreciate the thorough reading, apologies for all typos. We have carefully re-read and corrected the manuscript.

2. The feasibility of electrophoretic deposition-bubble collection technology needs to be validated. For the removal of nanoplastics from waters, the impacts of size (from tens to hundreds of nanometers), shapes (fiber, sphere, rod, irregularly-shaped debris) and surface-hydrophobicity can significantly change the capture and separation behaviors. Typically, the aggregation and dispersity of PMMA nanoplastics in water or wastewater are still unknown before treatment.

We fully agree with Reviewer 4 that all of these points are relevant. In the revised version, we have taken care to address systematically as many of the points as mentioned. To this end, we have first included model systems of spherical particles with different surface chemistry and material composition, to fundamentally evaluate the mechanism and removal efficiency. The amended section now reads:

P 10: We subjected a range of different model dispersions to the ePhoam process to probe the versatility of the method, in particular with respect to the material and the surface properties of the dispersions. To this end, we synthesized polystyrene (PS) colloidal particles with different surface

functionalities by changing the comonomer in the surfactant-free emulsion polymerization process^{67,68} (Figure 4). Carboxylic acid-functionalized PS nanoplastic dispersions showed a similar behavior compared to the PMMA-based model system, and exhibited recovery efficiencies of 96% after 5 operation cycles, both by anodic deposit and particle-stabilized foam (Figure 4a). Sulfonate-functionalized PS nanoplastic particles, which also have a negative surface charge (Figure S9), showed a comparable recovery efficiency with 92% total recovery after 5 cycles (Figure 4b). In this case, the initial recovery at the first operation cycles was lower, and a higher voltage (70 V) was required. These differences may be caused by the more hydrophilic nature of the sulfonate group that requires higher proton concentration in the vicinity of the anode to efficiently form a particle-stabilized foam (Figure S9). Positively-charged, amidine-functionalized PS nanoplastic particles also exhibited a pH-dependent change in surface properties, albeit with a decrease towards increasing pH values as the amidine groups become neutralized by deprotonation (Figure S9). In this case, the separation achieved comparable yields but the process was inverted, with the removal occurring at the cathode (70 V) in the form of deposited films and a particle-stabilized foam formed from the evolving hydrogen (Figure 4c). Finally, we probed a third material class by synthesizing poly(butylmethacrylate) (PBMA) colloidal dispersions with carboxylic acid surface functionalities⁶⁹. In this case, the surface properties are similar to the initial PMMA-based dispersion and both materials are methacrylates, but differ in their glass transition temperature (T_g). While PMMA has a T_g of ~ 105 °C, the T_g of PBMA is close to room temperature ($T_g \sim 33$ °C)⁷⁰, making the particles prone to film formation during operation. As a result, upon the separation, this dispersion only formed an unstable foam and the deposit on the anode was rather transparent. Both observations relate to the low T_g , which causes the particles to sinter during the process, leading to a homogeneous film on the anode and a mechanically less stable foam. Consequently, the recovery efficiency was lower and reached 63% after 5 cycles (Figure 4d). Noteworthily, this dispersion had a higher conductivity, and thus required a lower voltage (32 V) for the separation process. These experiments demonstrate that the separation process can be applied to a range of different materials and surface properties, but requires adapting of the process conditions to the specific characteristics (surface charge, T_g , conductivity).

Figure 4: Recovery of model nanoplastic particles with different material and surface properties by the electrophoretic deposition/ particle stabilized foam formation process. a) Polystyrene-carboxylate particles; b) Polystyrene-sulfonate particles; c) Polystyrene-amidine particles, d) Polybutylmethacrylate (PBMA) particles.

Secondly, we have investigated the influence of less defined particle systems (new Figure 6), and, in particular very ill-defined particles arising from eye glass and contact lens polish. These also show significant removal by our process, although the dispersion contains a range of particles of different material, shape, and size, in addition to various molecular additives. We have added an extensive characterization of these systems, including an evaluation of particle size and shape:

Figure S12: Particle size and shape characterization of the wastewater generated in the eye glass and contact lens polishing. a-f) SEM images of the particles found in the wastewater, showing different shapes and structures; g,h) SEM images of differently-sized spherical particles. i) Particle size distribution measured by Partica LA-950V2 laser diffraction particle size distribution analyzer.

Finally, we note that for the case of PMMA from industrial wastewater, the process works very well and allows removing almost all particles. By virtue of the added surfactant within the production process, these dispersions are colloidally stable and not agglomerated. Notably, as we show in the revised version (Supplementary Figure 11), the removed particles could be recycled for the process since they did not show any signs of degradation.

3. For the nanoplastics-polluted wastewater, the interaction of organic matter molecules and nanoplastics influence the performance of treatment efficiency. However, it was not studied in this study.
4. The formation of nanoplastic foams by oxygen microbubbles can be impacted by water chemistry. For instance, the co-existing organic matter and particulate pollutants as well as the change of ionic strength should be investigated for their impact on nanoplastic-foams and thus the efficiency of nanoplastic removal.

In the revised version, we subject different industrial wastewaters to our separation process. In all cases, significant removal efficiencies were achieved, despite a range of different organic additives present in these wastewaters (see Figure 6). This clearly demonstrates the ability of the separation

process to remove nanoplastics from real wastewater, especially in industrial cleaning and recycling procedures. In particular, using the wastewater from the eye glass polishing industry, we probed the presence of a range of both organic and particulate pollutant. A systematic study on the detailed influence of individual, well controlled organic additives on both the electrochemistry and the foam formation process should be performed when focusing on distinct removal scenarios to narrow the parameter space. We hope that our study showcases the potential and will trigger precisely these studies towards optimization of the technology for individual separation scenarios.

5. The cost-efficiency analysis should be made and compared with traditional electrochemical and flotation treatment such that the advantages of the technology proposed in this work can be clarified.

We thank reviewer 4 for bringing up this important point, which clearly is relevant for a practical applicability of the process. We have performed a comprehensive analysis of the energy consumption per unit volume and compared the associated costs and carbon footprints with the conventional treatment of industrial wastewater from PMMA production (incineration). These estimations indicate that the process is economically viable. The relevant sections of the main text now reads:

P13: Compared to other electrochemical-driven technologies (Supplementary Table 3) and incineration, which is the current technological treatment of industrial wastewater from emulsion polymerization-based processes, the ePhoam process is economically viable and cheaper, in particular when factoring in the ability to recycle the polymer. Additionally, it combines attractive properties of other electrochemical processes (details in Supplementary Information). Implementing the electrophoretic deposition/particle stabilized foam formation process would lower the carbon footprint by approximately 77 % compared to the state-of-the-art waste treatment (details in Supplementary Information).

Two complete chapters on the energy estimation and cost of the process, its comparison to the currently applied technology for wastewater treatment, and the results on the carbon footprint are included in the Supplementary Information (Chapter 1.3 and 1.4).

REVIEWER COMMENTS

Reviewer #1 (Remarks to the Author):

I commend the author's efforts in enhancing the manuscript. The novelty of the study is now evident, and the discussion has achieved a more balanced perspective. Based on this, I recommend the publication of this work.

Reviewer #2 only provided confidential remarks to the editor.

Reviewer #3 (Remarks to the Author):

The authors have made improvements and more experimental details have been supplemented. However, almost all my main concerns have not been addressed.

1. In the response to my first question, the authors declared that “we underline that our process is unique in the combination of deposition and the in-situ generation of particle-stabilized foam, which arises without the requirement of any additional additives”. I strongly disagree with this statement. The so-called “in-situ generation of particle-stabilized foam” is merely a process of the electroflotation, which has been widely used for nanoplastics/micropastics removal. Thus, the method used in this study is not a novel technique, and it is inappropriate to define it as a new method. In addition, compared with current electrochemical methods for nanoplastics/micropastics removal, this method does not show any interesting breakthroughs, in terms of the energy utilization efficiency and the removal performance.

2. For my second question, the authors’ response is irrelevant to my question. It is necessary to investigate the role of generated H₂ and O₂ on the nanoplastic removal process. The contribution of these gases to the nanoplastic removal performance should be quantitatively studied.

3. Also, the response to the third question is unacceptable. The stability of the formed foam can be measured in a quantitative way. The authors may study related techniques from previous reports on mineral froth flotation.

4. In the response to my fourth question, the authors said that they optimized the voltage formed for bubble formation. However, I have not seen related contents in the manuscript. In addition, how did the authors optimize the voltage? Which indexes are considered? Bubble size, amount, speed, or stability? Authors should provide sufficient information to support their statements.

5. To investigate the possible chemical changes of nanoplastics, it is necessary to analyze the recovered plastics’ surface chemical properties with FTIR, XPS; also, the treated water should be analyzed.

6. In the response to my ninth question authors have performed removal tests in real wastewater, it is interesting. However, are there any organic/inorganic pollutants in the wastewater? How did these pollutants affect the removal of nanoplastics?

7. In the response to my tenth question, the author stated that “the ePhoam process is economically viable and cheaper, in particular when factoring in the ability to recycle the polymer”. This is highly unacceptable compared to current studies, in terms of the energy costs, carbon footprint, and the recycled plastics. In most studies, plastics are recovered with a high stability, and all of them can be used for recycling the polymers.

8. In the removal of other nanoplastics (e.g., PBMA), the method shows quite low efficiency. The authors said it was due to the formation of film on electrodes, which indicates a serious drawback of this method. Thus, this method may not be suitable for practical applications which generally contains mixed plastics.

9. As stable foams formed on the electrodes, the active surface of electrodes will be blocked and the performance might be weakened, which would lead to severe stability issue, while the authors did not consider this. Are there any practical methods to address this issue.

Reviewer #4 (Remarks to the Author):

General comments:

I appreciate the work that the authors supplemented to address my comments given in the 1st revision. Indeed, the technical novelty of this work is not high. Moreover, the authors are suggested to design the experiments and discuss the results from the viewpoints of both scientific and practical aspects in the domain of environmental engineering. Some specific comments are provided down below:

Specific comments:

1. The section of INTRODUCTION could be well improved via demonstrating the technical defects and limitation of wastewater treatment technologies in nanoplastic removal. Some conventional technologies have been listed in this section after the 1st revision of manuscript but it is suggested to focus on the interfacial processes during treatment as well as the removal of nanoscale plastics rather than micron plastics.
2. No standard deviation was presented in Fig. 1, Fig. S2 or table in SI-1.2 (COD and TOC measurement protocol and Results). The repeats of experiments (not measurements) were not mentioned in the manuscript.
3. A fatal issue of the present work might be the concentration of nanoplastic particles in the simulating wastewaters. The authors did not provide any information about the real concentrations of nanoplastics in the practical scenarios. The reviewer made a rough calculation according to the data given in the paper, and the nanoplastic concentration could be 10^{16} particles/L or even higher. The pollutant loading is too heavy when the proposed technology is adopted as wastewater treatment method, and the pretreatment process is needed in the real wastewater treatment practice.

Reviewer #1 (Remarks to the Author):

I commend the author's efforts in enhancing the manuscript. The novelty of the study is now evident, and the discussion has achieved a more balanced perspective. Based on this, I recommend the publication of this work.

Reviewer #2 only provided confidential remarks to the editor.

Reviewer #3

#1. In the response to my first question, the authors declared that “we underline that our process is unique in the combination of deposition and the in-situ generation of particle-stabilized foam, which arises without the requirement of any additional additives”.

I strongly disagree with this statement. The so-called “in-situ generation of particle-stabilized foam” is merely a process of the electroflotation, which has been widely used for nanoplastics/microplastics removal.

Thus, the method used in this study is not a novel technique, and it is impropriated to define it as a new method. In addition, compared with current electrochemical methods for nanoplastics/microplastics removal, this method does not show any interesting breakthroughs, in terms of the energy utilization efficiency and the removal performance.

We apologize if there was a misunderstanding in our answer in the first revision. Rev. 3 is absolutely correct that there are similarities of our process to conventional electroflotation. Most importantly, both exploit gas bubbles produced by water splitting for particle capture. However, there are also important differences. Electroflotation uses water splitting as an (efficient) means to generate bubbles within the dispersion – in contrast to other flotation techniques where the bubbles enter the liquid externally. The adsorption process occurs in the volume of the liquid, and the foam can subsequently be removed from the top. This is why this process is very efficient in capturing comparably hydrophobic particles¹, and often relies on sacrificial electrodes to destabilize the dispersion by releasing multivalent ions to increase efficiency²⁻⁴. In our process, the electrode and its local environment play a much more decisive role. In this case, we i) exploit phoretic movement of particles to the electrode surface to enhance capture efficiency; and ii) efficiently exploit the local change in pH to directly manipulate the hydrophilicity of the particles and thus facilitate foam formation. As a consequence, this process is efficient at capturing hydrophilic particles in colloidal stable dispersions without requiring the manipulation of the bulk phase (as is the case for sacrificial electrodes that operate by ion release). Secondly, the particle-stabilized foam is produced directly on the electrode surface (in electroflotation it is separated from the electrode), which allows the design of roller systems for continuous processing and scale up. A second important difference is the yield in our process, which increases with increasing particle load in the wastewater (see Figure 7), while it decreases in electroflotation¹. As such, while both processes share a similar basis, we believe that these differences make them complementary and amendable to different treatment scenarios.

To illustrate these differences, we directly compare electroflotation and the ePhoam process in terms of the formed foam, its stability, and deposit on the electrode surface. To this end, we performed the electroflotation experiment using the same electrodes material DSA but in a different reactor and setup as conventionally used in electroflotation. As shown in the new Figure S5, the formed foam floated at the surface of the electroflotation reactor but was not observed on the electrode surface. Thus, the local pH change is not the dominating mechanism, and the stability of the foam was very low because of the hydrophilic nature of the particles. This is further supported by local pH measurements at the dispersion/air interface (pink box in Figure S5 a), which was similar to the pH of the bulk dispersion (pH=2.4). Hence, the particles conserve their hydrophilic properties and do not form a stable foam.

In the revised manuscript, we have extended the description of additional methods and explicitly introduced and discussed electroflotation. We have then added a section underlining the differences to our process based on the described experiments:

- P.2/3: Electroflotation removes hydrophobic particles with nascent bubbles emerging from electrodes undergoing gas evolution by water splitting^{26,27}. As a relevant example in the context of plastic pollution, this process has been successfully employed in wastewater generated from solvent-based automotive paints, where the plastic pollution forms by washing steps²⁷. An additional contribution to efficient capture of particles can be the release of multivalent ions when sacrificial electrodes are used. These ions destabilize colloidal systems and thus facilitate their capture by the bubbles^{26,28,29}. However, the separation efficiency decreases with increasing particle load in the wastewater^{27,30}, and the removal of dispersed, colloiddally stable particles is inefficient due to the low propensity of adsorption to the bubble^{31,32}.
- P. 6: Unlike conventional electroflotation, the particle-stabilized foam forms directly at the electrode, allowing for the exploitation of both the phoretic movement of particles toward the nascent bubbles and of the local pH changes to enhance the interfacial adsorption efficiency. In contrast, in electroflotation, the adsorption process of particles to the bubble occurs within the dispersion volume and thus cannot exploit these effects. As a result, conventional electroflotation is less efficient in separating nanoplastics from colloiddally stable dispersions as it does not generate a stable foam. (See Supplementary Figure S5 and Supplementary Movie 2.)

Figure S5. Comparison to conventional electroflotation. **(a)** Setup for electroflotation experiment using the PMMA model system and the mixed metal oxides electrodes (DSA). Note the limited amount of observed foam, indicating instable foam formation. The pH in the boxed area was similar to that of the bulk solution (pH=2.4) **(b)** ePhoam process. Stable foam on the anode surface in the roller setup being collected in the doctor blade part. **(c)** Stable foam formed on the anode surface in the discontinuous setup.

Secondly, in response to the criticism of usefulness of our process, we emphasize that despite the progress and development of other processes (see comparison in Tables S3 and S4 for both electrochemical processes and conventional non electrochemical processes) to this point, the separation of colloiddally stable particles from wastewater, especially in high concentrations as occurring in industrial wastewaters remains a challenge. Electroflotation as a case in point has been shown to work e.g. for particles formed in automotive coatings processes¹, but i) these particles are hydrophobic in nature (because they arise from solvent-based coatings) and thus more susceptible to interfacial adsorption, and ii) the process yield decreases with increasing particle load. We show that our process efficiently removes colloiddally stable particles in three real environmental wastewaters as PMMA industrial wastewater, the water-based paint and the eyeglass and contact lenses polishing wastewater, in addition to several model particles with different charges and functional groups. In terms of cost efficiency and footprint, it is difficult to accurately compare the processes as they are efficient in different scenarios. Hence, we have added a comparison to the current process of industrial PMMA wastewater treatment – the same wastewater we subject to our process.

#2. For my second question, the authors' response is irrelevant to my question. It is necessary to investigate the role of generated H₂ and O₂ on the nanoplastic removal process. The contribution of these gases to the nanoplastic removal performance should be quantitatively studied.

To confirm our previous statement that the formed foam on the vicinity of the anode is mainly composed of O₂ gas bubbles we performed two sets of experiments. Firstly, an analytical characterization of the formed foam through the EGA technique (Evolved gas analysis) which is based on Thermogravimetric analysis and Mass spectrometry (TGA-MS). The escaping gases are detected as a function of temperature. In the case of mixtures of a few substances escaping at the same time, MS mass traces help to assign individual components. In addition, smaller molecules (e.g., water, oxygen, nitrogen, carbon monoxide, etc.) and changes in the gas atmosphere could also be recorded. The results of the Evolved gas analysis through TGA-MS are depicted in Figure S4, which shows the detection of O₂, N₂, CO₂ and H₂. This analysis shows that oxygen is much overrepresented in the sample compared to what would be expected if the bubbles contained air. This supports the hypothesis that the oxygen evolution reaction caused the stable bubbles. Note that due to gas exchange in the porous foams between production and measurement, also nitrogen and CO₂ is detected.

Secondly, we directly assessed which gas forms the stable foam by selectively separating the gas evolution from the colloidal dispersion via a nylon membrane with a porosity of 0.1 μm, which prevents the contact of particles and bubbles (both cannot pass the membrane). Figure S4 shows that when the H₂ gas bubbles are blocked at the cathode side, we still observe a stable foam on the anode surface (left side). In contrast, when the O₂ gas bubbles are blocked by the nylon membrane, the particles deposit on the anode surface but without any foam formation, just as deposition in a form of film, while the cathode remains free of bubbles. This experiment demonstrate that it is indeed the oxygen bubbles forming at the anode that cause the foam formation (in the negatively charge model system). We have added and discussed the data in the manuscript and SI:

P.6: We further use mass spectroscopy and selective blocking of individual electrodes to confirm that the stable foam indeed forms from oxygen bubbles at the anode (Figure S4).

Figure S4. a) Individual mass traces plotted over temperature based on EGA analysis. b, c) Photos of both electrodes after the separation process using a Nylon membrane (0.1 μm) b) firstly covering the cathode and c) secondly covering the anode.

#3. Also, the response to the third question is unacceptable. The stability of the formed foam can be measured in a quantitative way. The authors may study related techniques from previous reports on mineral froth flotation.

In the revised version, we have now quantified the foam stability via the change in foam height as a function of time after switching of the electric field. As can be seen in the new Figure S8, the foam remains stable over the period under investigation. In fact, even after complete drying, a solid foam with permanently fixed structure resulted. We further investigated the impact of the voltage on the foam formation. As can be seen, with increasing voltage a higher initial foam volume is formed – as expected from the increased amounts of generated bubbles. While all foams consolidated into a stable state (evidenced by the plateaus at later times), the foam showed different initial dynamics: With increasing voltage, the initial height was reduced. We hypothesize that this is related to both the increased bubble formation, which may generate interfaces faster than the particles can attach, and the increase in temperature, which lowers surface tension and thus reduces the interfacial adsorption strength (the temperature indeed increases). Note that we are not able to experimentally test these hypotheses. We have amended the manuscript as follows:

- P6: ...the formation of a persistent, particle-stabilized foam at the anode (Figure 1g) The foam, which can be effortlessly extracted from the dispersion (see Figure 1a and Figure S3), remains stable for an extended period of time (Supplementary Movie 1).
- P8: The foam amount and stability changed as a function of applied voltage (Figure S8), with the foam volume increasing with the voltage, as expected from the Faraday's law of electrolysis⁷⁰. Above the threshold voltage for bubble formation, all voltages produced a foam with long-term stability, although the initial dynamics changed. The highest voltage showed the strongest decrease in foam volume before reaching a stable state in the measurement cell, despite yielding the largest foam volume. We hypothesize that this may be related to the increased bubble formation rate, possibly generating interfaces faster than the particles can attach to, and the increase in temperature lowering surface tension. In the following, we fixed the voltage to 50 V as this produced the highest total recovery.

Figure S8: a) Photos of the stable foam standing on the anode surface inside the electrolytic cell filmed for 2 hours and captured as photos every 10 minutes at different voltages (V= 50 V in this figure). The produced foam is highlighted with arrows. b) Foam samples at i) room temperature after 24 hours, and ii) dried stable foam after one week. c) Particle stabilized foam stability assessment through the foam height versus time at different voltages.

#4. In the response to my fourth question, the authors said that they optimized the voltage formed for bubble formation. However, I have not seen related contents in the manuscript. In addition, how did the authors optimize the voltage? Which indexes are considered? Bubble size, amount, speed, or stability? Authors should provide sufficient information to support their statements.

The applied voltage has a significant impact on both sub-mechanisms of the ePhoam process: The electrophoretic deposition and the particle stabilized foam. The influence of the applied voltage depends on the target application of these mechanisms, which is in our case the recovery percentage of the nano plastics from the wastewater. That is why we have assessed the recovery percentage at different voltages.

If we assess each sub-mechanism separately, the impact of applied voltage could differ. For instance, in the case of the electrophoretic deposition, the increase of applied voltage helps to increase the electrophoretic mobility of the charged particles in the electric field and thus achieve a faster process. However, simultaneously, the high applied voltage can generate higher volume of the electrogenerated gas bubbles and thus disturb the deposit/ film formed on the electrode surface. Similarly, in the case of the particle stabilized foam, the increase of applied voltage can generate more foam due to the increase of gas bubbles from water electrolysis but it can also lead to an increasing instability of the foam as the bubbles form faster than the absorption process, and as the temperature increases (see discussion in point above).

In our case, we have chosen the voltage in a way that the mass of recovered particles was maximum (See Figure 2). This was the empirical reason for operating the process at 50V to evaluate the performance. Note that at this voltage the recovery seemingly converged, which is why we did not further increase the voltage. Thanks to the comment of Rev. 3 on the foam formation process, we were now able to provide a discussion/rationale behind this behavior. As can be seen in the new Figure S8, the total foam volume after consolidation was largest with 50V in our model system. Therefore, while 40V produced the most stable foam (note the small decrease in foam stability in this case), 50V had a higher initial loss in volume, but a higher total volume. This larger foam volume translates into a larger mass of recovered particles, as shown in Figure 2. We have now explicitly stated the reason for choosing 50V in the discussion about the foam stability:

P8: The highest voltage showed the strongest decrease in foam volume before reaching a stable state in the measurement cell, despite yielding the largest foam volume. [...] In the following, we fixed the voltage to 50V as this produced the highest total recovery.

#5. To investigate the possible chemical changes of nanoplastics, it is necessary to analyze the recovered plastics' surface chemical properties with FTIR, XPS; also, the treated water should be analyzed.

We thank the reviewer for this suggestion. In the revised manuscript, we have now extensively compared the thermal (Differential Scanning Calorimetry, DSC), chemical (IR Spectroscopy), Molecular weight (GPC), and surface properties (X-Ray Photoemission spectroscopy, XPS) of powders before and after treatment, using the most relevant example for large-scale application and recycling of PMMA wastewater. All measurements indicate that the separation process does not decompose or chemically alter the material. Molecular weight, thermal properties, chemical composition, as well as surface properties of the powder remain essentially unaffected by the separation process. We already analyzed the treated water in Figure 5 of the manuscript.

We have added all data to the SI and amended the manuscript as follows:

P. 13: To further confirm this recycling potential, we investigated the thermal properties, molecular weight distributions, chemical composition, as well as surface properties of the PMMA powder before and after separation (Figure S13-S16, Table S7). All measurements indicate that the separation process does not degrade or chemically alter the polymers, and thus indicate the possibility to recycle the recovered material.

Figure S14. Differential scanning calorimetry (DSC) characterization of the PMMA particles of industrial wastewater a) before and b) after the separation process.

Figure S15. Attenuated total reflectance infrared (ATR-IR) spectroscopy analyses of the PMMA particles of industrial wastewater before and after the separation process.

Figure S16. X-ray photoelectron spectrometry (XPS) of PMMA particles of industrial wastewater before and after the separation process a) Overlay plot of the C 1s- sum signals b) Overlay plot of the O 1s- sum signals.

#6. In the response to my ninth question authors have performed removal tests in real wastewater, it is interesting. However, are there any organic/inorganic pollutants in the wastewater? How did these pollutants affect the removal of nanoplastics?

Both eyeglass/ contact lenses polishing wastewater and wastewater from commercial dispersion paints indeed contain different organic and inorganic additives. These are often trade secrets of the company and encompass anything from pigments, via stabilizers to antimicrobial agents, all in unknown concentrations. For the eyeglass polishing wastewater, we performed an extensive analysis of the content, based on IR and mass spectroscopy (please see Figures S18, S19 and S20). For instance, Tinuvin (a UV absorber) and Bumetrizole was detected in our eye glass polishing wastewater, and it has been shown in other studies that it has a high bioaccumulation potential in aquatic food webs⁶. UV filters and stabilizers that are widely used in eyeglass industry are consequently discharged in the polishing wastewater, which creates additional pollutants other than nano and microplastics. Moreover,

in the dispersion paint, the manufacturer specifies that the paint contains anti-microbial compounds (2-Methyl-2H-isothiazol-3-one and 1,2-Benzisothiazol-3(2H)-one). Despite the presence of multiple organic/ inorganic pollutants in the real environmental wastewater, the ePhoam process efficiently separated the particles. As has been shown through the eight tested systems, each type of wastewater has a slightly different behavior in terms of recovery form (more stable foam in some cases, more film formation in others...) and each system required a slight adjustment of the process parameters. However, in all cases, suitable conditions that were able to remove significant parts of the nanoplastic contaminant could be identified. This, in our view, demonstrates the versatility and applicability of the method. How exactly the molecular pollutants affect the separation process is difficult to investigate because the same system is not available with and without such pollutants. More detailed studies on model systems with known amounts of common additives could help to address this question in detail (and resolve if, e.g. these pollutants are also oxidized by electrode reactions), but are not in the scope of this manuscript.

#7. In the response to my tenth question, the author stated that “the ePhoam process is economically viable and cheaper, in particular when factoring in the ability to recycle the polymer”. This is highly unacceptable compared to current studies, in terms of the energy costs, carbon footprint, and the recycled plastics. In most studies, plastics are recovered with a high stability, and all of them can be used for recycling the polymers.

As we argue above, there are differences in the capacities of our process compared to existing separation processes. In particular, the performance of our process in separating colloiddally stable, hydrophilic particle dispersion at high concentrations is unmatched by other processes. The challenge is evidenced by the fact that in PMMA production, incineration of wastewater remains the industrial solution to remove waste water, since apparently, no other technology has provided a viable alternative. We therefore base our comparison of cost and carbon footprint on the comparison of this state of the art and our process (since both treat exactly the same particle dispersion). Important in the context of recycling is also the fact that our process does not require the addition of any other substances, or the use of sacrificial electrodes that need replacement and release metal ions which affect the properties of the recycled product. Note that when we use normal stainless steel electrodes, iron ions are incorporated into the recovered material, evidenced by the change in color; see e.g. Figure 3). Such impurities require difficult down-stream purification or otherwise compromise product quality. Hence, a process that recovers materials without impurities and with essentially similar polymer characteristics (as we show in the new Figures S13-S16) is desirable.

#8. In the removal of other nanoplastics (e.g., PBMA), the method shows quite low efficiency. The authors said it was due to the formation of film on electrodes, which indicates a serious drawback of this method. Thus, this method may not be suitable for practical applications which generally contains mixed plastics.

We have shown an efficient separation of nanoplastics, far exceeding 90% recovery in the three distinct real life wastewater systems with vastly differing properties, as well as for four different model systems with different chemical compositions and surface properties. The process has shown to be scalable and can efficiently separate nanoplastic particles across a wide range of concentrations, including relevant scenarios of high concentrations as occurring in downstream processing in industrial wastewaters. The industrial waste waters include relevant examples of industrial raw materials (PMMA) and widely applied consumer products (dispersion paints), which are used on tremendous scales. In our view, this zoo of particle systems across different sectors, with different compositions, additives, mixtures, and concentration ranges gives evidence of practical applicability. Notably, these demonstrations include very well defined target scenarios, such as the downstream processing and recycling directly in the PMMA production. Singling out a system that is not as efficiently separated (albeit the recovery is above 60%) in our view does not generally disqualify the process, but rather serves as an assessment of boundary conditions of the process.

#9. As stable foams formed on the electrodes, the active surface of electrodes will be blocked and the performance might be weakened, which would lead to severe stability issue, while the authors did not consider this. Are there any practical methods to address this issue.

Rev. 3 correctly points out that blockage of the active area can reduce performance in the model setup using batch processes. To clarify this point, we have added an explanation that the foam was removed after each operation cycle to avoid misinterpretations:

P.9: As the DSA electrodes developed large amounts of foam in a short time, we resorted to a cyclic operation in which we assessed the particle recovery by the ePhoam process as a function of 4 min operation cycles, where we removed the foam accumulated on the electrode surface after each cycle

However, in the manuscript, we use the batch setup only for model studies to explore the mechanism in detail and compare different model systems. We introduce a pilot plant (Figure 7) that allows scale up because of the continuous operation and at the same time directly mitigates the issue of foam build-up on the electrode, as a doctor blade efficiently removes the foam and recovers the electrode. In the manuscript, we discussed this setup, but have now added a movie to further underline the separation:

P. 16: In the course of operation, the anode rotates, picks up particle deposits, both as foam and thin film, which is subsequently removed from the electrode by a doctor blade. As a result, the removed particles can be recycled, while the electrode surface is recovered and re-immersed (Supplementary Movie 6).

Rev4

General comments:

I appreciate the work that the authors supplemented to address my comments given in the 1st revision. Indeed, the technical novelty of this work is not high. Moreover, the authors are suggested to design the experiments and discuss the results from the viewpoints of both scientific and practical aspects in the domain of environmental engineering. Some specific comments are provided down below:

#1. The section of INTRODUCTION could be well improved via demonstrating the technical defects and limitation of wastewater treatment technologies in nanoplastic removal. Some conventional technologies have been listed in this section after the 1st revision of manuscript but it is suggested to focus on the interfacial processes during treatment as well as the removal of nanoscale plastics rather than micron plastics.

We thank the reviewer for this suggestion. In the revised version, we have further extended the description of conventional technologies and discussed their applicability and shortcomings in the context of nanoplastic removal. Furthermore, we have included a clearer description on how our process differs from these existing approaches, and why these differences allow us to separate materials that are challenging for existing technologies (such as colloidally stable, hydrophilic particles):

P.2/3: Electroflotation removes hydrophobic particles with nascent bubbles emerging from electrodes undergoing gas evolution by water splitting^{26,27}. As a relevant example in the context of plastic pollution, this process has been successfully employed in wastewater generated from solvent-based automotive paints, where the plastic pollution forms by washing steps²⁷. An additional contribution to efficient capture of particles can be the release of multivalent ions when sacrificial electrodes are used. These ions destabilize colloidal systems and thus facilitate their capture by the bubbles^{26,28,29}. However, the separation efficiency decreases with increasing

particle load in the wastewater^{27,30}, and the removal of dispersed, colloiddally stable particles is inefficient due to the low propensity of adsorption to the bubble^{31,32}.

P. 6: Unlike conventional electroflotation, the particle-stabilized foam forms directly at the electrode, allowing for the exploitation of both the phoretic movement of particles toward the nascent bubbles and of the local pH changes to enhance the interfacial adsorption efficiency. In contrast, in electroflotation, the adsorption process of particles to the bubble occurs within the dispersion volume and thus cannot exploit these effects. As a result, conventional electroflotation is less efficient in separating nanoplastics from colloiddally stable dispersions as it does not generate a stable foam. (See Supplementary Figure S5 and Supplementary Movie 2.)

#2. No standard deviation was presented in Fig. 1, Fig. S2 or table in SI-1.2 (COD and TOC measurement protocol and Results). The repeats of experiments (not measurments) were not mentioned in the manuscript.

All standard deviations have been added to the figures and tables and descriptions of the repetitions added to the experimental descriptions.

#3. A fatal issue of the present work might be the concentration of nanoplastic particles in the simulating wastewaters. The authors did not provide any information about the real concentrations of nanoplastics in the practical scenarios. The reviewer made a rough calculation according to the data given in the paper, and the nanoplastic concentration could be 1016 particles/L or even higher. The pollutant loading is too heavy when the proposed technology is adopted as wastewater treatment method, and the pretreatment process is needed in the real wastewater treatment practice.

The concentration of nanoplastic particles in the model system (simulating wastewater as mentioned by the reviewer) was tested from low to high (0.01 to 1 wt.%); see Figure S6. Our process has been shown to work for all range of concentrations. Notably, at lowest particle loads in the simulated wastewater, the removal efficiency seems to decrease, but has not been optimized. Longer process times or an adjusted setup (e.g smaller electrode distance) can increase the yield in these scenarios. In the manuscript, instead of focusing on lower limits for particle separation, we explore practical scenarios, with the concentration ranges typically encountered there. In the previous version, we discussed e.g. PMMA industrial wastewater, which contains a concentration of 5 to 7 wt.% of particles. Moreover, in our 2nd practical scenario, which is the commercial dispersion paint, we have tried to clean a small brush that was inside the paint bottle (commercially sold as ~30 wt.%) with tap water until it was clean and we ended up with wastewater of ~3 wt.%. Washing out paint brushes from dispersion paints, generate highly concentrated wastewater as well. In the case of eyeglass and contact lenses polishing wastewater, we have also mentioned in the last revision that the concentration of the solid content after filtration of bigger microparticles, was 0.03 wt.%. Yet, the separation process remained efficient. These examples show that the process is flexible and can be adopted to a wide range of particle concentrations in wastewater. Furthermore, in Figure 5 we characterize the cleaned wastewater after removal, both in terms of chemical oxygen demand and total organic carbon content. Both metrics show a drastic reduction after cleaning. In fact, the values obtained from this simple test system fall within the regulatory threshold for waste from the chemical industry, as specified by the Federal Ministry for the Environment, Nature Conservation and Nuclear Safety in Germany, albeit to our knowledge, no defined threshold values specific for nanoplastic emission exist. Note that the yield of removal of the process can be further increased by increasing treatment time, serializing multiple units, or decreasing the electrode distances. A detailed optimization and adaption for particular waste water scenarios is not in the scope of this manuscript, which introduces the general technique, discusses the mechanism, explores versatility and boundaries, and outlines potential fields of application.

References

1. Mohtashami, R. & Shang, J. Q. Treatment of automotive paint wastewater in continuous-flow electroflotation reactor. *Journal of Cleaner Production* **218**, 335–346; 10.1016/j.jclepro.2019.01.326 (2019).
2. Kyzas, G. Z. & Matis, K. A. Electroflotation process: A review. *Journal of Molecular Liquids* **220**, 657–664; 10.1016/j.molliq.2016.04.128 (2016).
3. Vu, T. P. *et al.* Characteristics of an electrocoagulation–electroflotation process in separating powdered activated carbon from urban wastewater effluent. *Separation and Purification Technology* **134**, 196–203; 10.1016/j.seppur.2014.07.038 (2014).
4. Baierle, F. *et al.* Biomass from microalgae separation by electroflotation with iron and aluminum spiral electrodes. *Chemical Engineering Journal* **267**, 274–281; 10.1016/j.cej.2015.01.031 (2015).
5. Chakik, F. e., Kaddami, M. & Mikou, M. Effect of operating parameters on hydrogen production by electrolysis of water. *International Journal of Hydrogen Energy* **42**, 25550–25557; 10.1016/j.ijhydene.2017.07.015 (2017).
6. Wang, W., Lee, I.-S. & Oh, J.-E. Specific-accumulation and trophic transfer of UV filters and stabilizers in marine food web. *Science of The Total Environment* **825**, 154079; 10.1016/j.scitotenv.2022.154079 (2022).

REVIEWER COMMENTS

Reviewer #3 (Remarks to the Author):

Some issues have been well addressed. However, following questions need to be further considered.

1. The answer to my 2nd question is unacceptable. The authors used nylon membrane with a porosity of 0.1 μm to separate the gas evolution from the colloidal dispersion. Authors should know that a large quantity of nano-sized bubbles will be produced during the process, how could the nylon membrane work? In this context, the gas bubble size should be measured at different voltages. In addition, the volume of H₂ should be twice that of O₂, why H₂ cannot play an important role in the removal process?
2. The measurement of the foam stability is unreliable. In this process, the measured data are the foam height, not the stability. A microscopic study of the foam generation, growth, and burst should be performed.
3. Authors need to calculate the cost of the process for treating per tone of wastewater, and compare with recent studies. Otherwise, how could authors the process is economically viable and cheaper? Actually, current adsorption, electro-flotation process have been widely used, does this process are more economically attractive or not?
4. XPS fine spectra of elements is required for analyzing the possible chemical change of plastics.

Reviewer #4 (Remarks to the Author):

The authors tried hard to reply all my comments though the inherent defects of this study cannot be well overcome or addressed. As a combination of electro-flotation and froth flotation, the proposed technology was not novel from the viewpoint of either technical advances or process mechanisms. There have been a number of similar studies published in the scientific journals in the domain of environmental science and ecology. The paper might be accepted, but if not, I would suggest it to be transferred to Scientific Reports or other journals focusing on separation technologies such as Separation and Purification Technology.

REVIEWER COMMENTS

Reviewer #3 (Remarks to the Author):

Some issues have been well addressed.

We are happy that reviewer 3 appreciated our efforts in the revisions and would like to thank her/him for the constructive feedback in the last round of revisions.

However, following questions need to be further considered.

1. The answer to my 2nd question is unacceptable. The authors used nylon membrane with a porosity of 0.1 μm to separate the gas evolution from the colloidal dispersion. Authors should know that a large quantity of nano-sized bubbles will be produced during the process, how could the nylon membrane work? In this context, the gas bubble size should be measured at different voltages. In addition, the volume of H₂ should be twice that of O₂, why H₂ cannot play an important role in the removal process?

In the initial question, the reviewer asked us to clarify which bubbles (i.e. hydrogen or oxygen) are causing the foaming. This is an important question that already came up in the first round of revisions and seems to persist. In the following, we are explaining the key steps of the process to resolve any possible misunderstandings:

- 1) Particles travel to different electrodes depending on their charge. If we use negatively charged particles, as is the case e.g. in our PMMA model system and the industrial PMMA dispersions, the particles inherently travel to the positively charged electrode, which is the anode. This causes them to be deposited there and increases the likelihood to be caught in the emergent bubbles as well.
- 2) More important in the context of stable foam formation is the process that occurs at this electrode. Due to the water splitting, at the anode side, the pH drops in the direct vicinity of the electrode (see Figure 1). This, in turn, renders the particles more hydrophobic, since their surface charge is reduced (also see Figure 1). This is why the particles can be captured by and adsorb to the formed bubble only in the vicinity of the electrode. In the case of our model system, these bubbles are made from oxygen, because oxygen evolves at the anode. The same particles that get in contact with other gas bubbles (evolving hydrogen bubbles at the cathode, see Fig. S4, or simply gas bubbled through the dispersion away from the electrodes (Fig. 1f)) will not efficiently adsorb to the bubble and thus do not form a stable foam, because the necessary local change in pH, which changes the hydrophilicity of the particles is not available. In summary, for negatively charged particles, only oxygen bubbles at the anode cause stable foam formation.
- 3) The necessity of the local change in pH renders the adsorption to oxygen bubbles away from the electrode unattainable. This is the crucial difference to conventional electroflotation, where the electrodes only occupies a minute volume of the fluid cell (typically at the bottom) and the bubbles catch particles on their way to the surface of the liquid. In our colloidally stable dispersions, this process does not lead to stable foam formation, as we show by reference measurements (see Figure S5).
- 4) If a particle dispersion with a positive charge is used, it will travel to and be deposited at the cathode side. In this case, it is the hydrogen bubbles that capture the particles. Associated with this process is an *increase in pH*, which leads to *deprotonation* of positive charges (note that

such positive surface charges require a different functional group at the surface compared to the acrylate model systems) and thus a similar decrease in surface charge (see Fig 4c).

In summary, the nature of the gas bubble that is “active” in the process depends on the charge, and thus the surface chemistry of the particles. Positively charged particles can make use of the hydrogen bubbles (which, as Rev. 1 correctly points out, occur in twice the volume), negatively charged particles rely on the local environment of the anode and thus need to adsorb to the evolving oxygen bubbles.

In our opinion, we have elucidated this process in much detail, ranging from the local characterization of the pH of the electrode, coupled with measurements of surface charge (Figure 1), control measurements supporting the importance of the local changes in surface charge (Figure 1, Figure S5), measurements in dependence of the surface functionality (and thus charge) of the particles (Figure 4), and a determination of oxygen-enriched atmosphere in the final foam when produced by negatively charged particles (Figure S4).

The use of the membrane was simply an idea to further change the setting to favor one foam formation over the other. The idea was to block access of the colloidal particles to the electrode surface. The particles have sizes above 200nm and can thus not pass through the pores. At the same time, the membrane needs to be porous to provide access of the water phase to the electrode and thus allow water splitting to take place (this is why a complete coating would not work). The goal of this experiment is to show what happens at the side *not* covered by the membrane. Here, Figure S4 clearly shows that when the anode is covered, no foam occurs, neither at the anode, nor at the cathode. When the anode is free, however, large amounts of foam can be seen. Whether smaller amounts of nanobubbles can pass through the membrane is not directly relevant here (and a lot of gas will pass through the gap between membrane and electrode anyways), what matters is the direct comparison of the unblocked electrodes, which shows where the foam emerges. This is also why it is not in the scope of the manuscript to investigate the size of the bubbles in this case either.

We have revised the manuscript once again to clarify the different steps of the process and the requirements to form a stable foam. The following passages have been amended:

Abstract:

Here, we describe a process to remove nanoplastics from wastewater, which synergistically combines electrophoretic deposition processes and the formation of stable, particle-stabilized foams enabled by local changes in particle hydrophilicity triggered by the water electrolysis reaction at the electrode surface [...] Both model systems and industrial wastewater were efficiently treated and cleaned from nanoparticles with removal efficiencies exceeding 90%. The exploitation of local pH changes to greatly enhance particle adsorption to nascent bubbles directly at the vicinity of the electrode distinguishes our process from conventional electroflotation and allows separation of colloiddally stable nanoparticles from aqueous dispersions.

Introduction:

P2: Electroflotation removes hydrophobic particles by adsorption to bubbles generated at electrodes undergoing gas evolution by water splitting at the bottom of the reactor, which rise through the dispersion^{26,27}. [...] However, [...] the removal of dispersed, colloiddally stable particles is inefficient due to the low adsorption energy of such hydrophilic particles to the bubble surface^{31,32}.

P3: The spontaneous adsorption of colloidal particles to interfaces has long been used to form highly stable emulsions, termed Pickering emulsions^{45,46}, and foams, known as particle-stabilized foams^{47,48}. Conventional, colloiddally stable particles dispersed in an aqueous medium typically do not form such stable foams. To efficiently create particle stabilized foams, the surface chemistry of colloidal particles

needs to be carefully adjusted in line with the contact-angle dependent attachment strength, typically by controlling the pH^{49–52} or by introducing tailored functional groups to the particle surface⁴⁸.

P4: Thus, this process synergistically combines gas evolution as a typical limitation of EPD processes with the formation of a particle-stabilized foam enabled by the local changes in pH in the direct vicinity of the emerging gas bubble at the electrode surface. These processes collude to efficiently remove nanoplastic particles from an aqueous dispersion.

Results

P6: In the absence of an applied voltage (and thus, water splitting reactions and local pH change at the anode), gas bubbles can be externally added by bubbling oxygen into the particle dispersion, resembling the strategy of a floatation process. However, no foam formation was observed (Figure 1f), which is attributed to the hydrophilic character of typical colloidal dispersions prepared by emulsion polymerization. Consequently, destabilization of the particles must be ensured for foam formation. Only during water electrolysis, the pH reduction in the vicinity of the emergent oxygen bubble at the anode altered the surface properties of the particles in situ without the need of any additives, and increased their adhesion strength to the gas/water interface^{31,32,42,62}. This local pH change is the key for the formation of a persistent, particle-stabilized foam at the anode (Figure 1g).

P.6: We further use mass spectrometry and selective blocking of individual electrodes to confirm that the stable foam indeed forms from oxygen bubbles at the anode (Figure S4). The lack of detected hydrogen confirms that hydrogen bubbles forming at the cathode do not contribute to particle-stabilized foam as the increased pH at the electrode does not reduce the absolute surface charge of the particles (Figure 1d) and thus does not increase the propensity to form a stable foam.

P12: Positively-charged, amidine-functionalized PS nanoplastic particles also exhibited a pH-dependent change in surface properties, albeit with a decrease in surface charges towards higher pH values as the amidine groups become neutralized by deprotonation (Figure S11). In this case, the separation achieved comparable yields but the process was inverted, with the removal occurring at the cathode (70 V) in the form of deposited films and a particle-stabilized foam formed from the adsorption of particles to the evolving hydrogen bubbles enabled by the increased pH (Figure 4c)

Supporting Information:

Figure S4. a) Individual mass traces plotted over temperature based on EGA analysis. b, c) Photos of both electrodes after the separation process using a nylon membrane (0.1 μm) b) firstly covering the cathode and c) secondly covering the anode. The membrane blocks access of the colloidal particles to the electrode surface and thus to the local environment of the pH change. In addition, gas evolution mainly occurred at the electrode/membrane interface, and thus in absence of the colloidal particles. When the anode is covered, no foam appears, neither at the anode, nor at the cathode (b). When the anode is uncovered, however, large amounts of foam can be seen (c).

2. The measurement of the foam stability is unreliable. In this process, the measured data are the foam height, not the stability. A microscopic study of the foam generation, growth, and burst should be performed.

We agree with the reviewer that the foam formation process can be studied in much more detail, including microscopic analyses, dynamics, coarsening mechanisms, etc. Such studies are interesting, but will rely on many different parameters, including not only the voltage (as remarked by the reviewer), but also the composition of the aqueous phase (electrolyte concentration, additives), the chemical nature of the particles, their surface charge density, chemical nature of the surface groups, their size, concentration, and potentially also bulk properties of the polymers (such as their glass transition temperature). While such studies are definitely interesting from a fundamental point of view, we believe that they do not fall within the scope of the manuscript. In the manuscript, we introduce the process, establish the physics and mechanistic details underlying the separation, and explore scope, scalability, and limitations of the process. In this context, it is our belief that the macroscopic foam characterization via its height sufficiently demonstrates foam stability. Also note that details of the foam dynamics are not of immediate technological relevance for the separation, since in the developed setup, the evolving foam is continuously separated from the electrode (see video 6), and hence, does not need to be stable over longer periods of time.

3. Authors need to calculate the cost of the process for treating per tone of wastewater, and compare with recent studies. Otherwise, how could authors the process is economically viable and cheaper? Actually, current adsorption, electro-flotation process have been widely used, does this process are more economically attractive or not?

Supplementary Table S3 compared our process to other technologies. We note, however, that for the most likely business case that we have identified, which is the separation and recycling of PMMA nanoparticles from industrial wastewater, no efficient separation technique exists to date. As our reference measurements show (Figure S5), conventional electroflotation does not produce significant foam and fails to efficiently remove these colloidally stable particles. The underlying reason for this challenge is that electroflotation relies on the spontaneous adsorption of particles to bubbles that pass through the water phase. While this works well for comparably hydrophobic particles (examples exist, e.g. for particles from polymer coatings, see Ref. 27), it fails for colloidally stable particles that do not efficiently adsorb to the interface because of their low contact angle (see, e.g. Ref. 31,32). In contrast, our process exploits local changes in surface hydrophilicity generated at the vicinities of the electrodes (see description to point 1).

We therefore base our assessment of economic and ecologic viability on the comparison to the standard procedure of wastewater treatment, which is the incineration of wastewater. The details of these calculations are described in the Supporting Information and are copied below for convenience. We have now also added a Figure to graphically compare the two processes (new Figure S17) and have added a clearer discussion in the main text body:

P. 13: Compared to other electrochemical-driven technologies (Supplementary Table 3) and incineration, which is the current technological treatment of industrial wastewater from emulsion polymerization-based processes, the ePhoam process is economically viable and cheaper, in particular when factoring in the ability to recycle the polymer. A direct comparison to incineration, using energy costs of July 2023, factoring in estimated operational- capital costs and payback for recycled polymer, and assuming a continuous process (see below) shows a cost reduction of 89% and savings of ~178 Euros per ton of wastewater (Figure S17). In addition, adoption of the ePhoam process in wastewater

treatment at such industrial emulsion polymerization facilities would lower the carbon footprint by approximately 77% compared to the state-of-the-art waste treatment of incineration (details in Supplementary Information, Figure S17).

Supplementary Information: cost calculation:

- The power consumption for the process is calculated by multiplying the voltage and the current.

$$20 V * 20 A = 400 W = 0.4 kW$$

- The process cleans 5 kg of wastewater (ww) in 2 h which leads to a starting mass flow of 2.5 kg/h.

$$\frac{5 \text{ kg ww}}{2 \text{ h}} = 2.5 \frac{\text{kg ww}}{\text{h}}$$

- By dividing the power consumption by the start mass flow the energy consumption per mass flow or per kg of wastewater is calculated.

$$\frac{0.4 \text{ kW}}{2.5 \frac{\text{kg ww}}{\text{h}}} = 0.16 \frac{\text{kWh}}{\text{kg ww}}$$

- The current cost of energy (July 2023) is 0.12 €/kWh so the cost of the energy consumption for the process is 19.2 €/t ww.

$$0.16 \frac{\text{kWh}}{\text{kg ww}} * 0.12 \frac{\text{€}}{\text{kWh}} = 0.0192 \frac{\text{€}}{\text{kg ww}} = 19.2 \frac{\text{€}}{\text{t ww}}$$

- The operational expenses (opex) for drying, pumps, cooling, etc. are estimated with 10% of the energy costs according to experience.

$$19.2 \frac{\text{€}}{\text{t ww}} * 0.1 = 1.92 \frac{\text{€}}{\text{t ww}}$$

- For the capital expenditures (capex) of a 10 t ww/h plant a mass flow of 80,000 t/year is calculated. A year is estimated by 8,000 h.

$$10 \frac{\text{t}}{\text{h}} * 8,000 \frac{\text{h}}{\text{year}} = 80,000 \frac{\text{t}}{\text{year}}$$

- The estimated invest for a 10 t ww/h plant is 3,000,000 € based on experience with a small package unit. With 10 years depreciation time 300,000 €/year are calculated.

$$\frac{3,000,000 \text{ €}}{10 \text{ years}} = 300,000 \frac{\text{€}}{\text{year}}$$

- Per metric ton of wastewater, the capex costs calculate to 3.75 €/t ww.

$$\frac{300,000 \frac{\text{€}}{\text{year}}}{80,000 \frac{\text{t ww}}{\text{year}}} = 3.75 \frac{\text{€}}{\text{t ww}}$$

- The recovered mass of polymer for the 10 t/h plant can be calculated by multiplying the mass flow with the starting concentration of polymer in the wastewater which is 1 wt.% and the

recovery rate of 91 %.

$$10 \frac{t \text{ ww}}{h} * 0.01 \frac{t \text{ polymer}}{t \text{ ww}} * 0.91 = 0.091 \frac{t \text{ polymer}}{h}$$

$$\frac{0.091 \frac{t \text{ polymer}}{h}}{10 \frac{t \text{ ww}}{h}} = 0.0091 \frac{t \text{ polymer}}{t \text{ ww}}$$

- With a payback for the polymer of 400 €/t a payback of 3.64 € per ton of wastewater is generated.

$$0.0091 \frac{t \text{ polymer}}{t \text{ ww}} * 400 \text{ €} = 3.64 \text{ €}$$

- Combined the costs for energy consumption, capex, opex with the payback for the polymer sum up to 21.23 €/t ww.

$$19.2 \frac{\text{€}}{t \text{ ww}} + 1.92 \frac{\text{€}}{t \text{ ww}} + 3.75 \frac{\text{€}}{t \text{ ww}} - 3.64 \frac{\text{€}}{t \text{ ww}} = 21.23 \frac{\text{€}}{t \text{ ww}}$$

- Compared to the alternative process that is used now (incineration of wastewater) and costs 200 €/t ww 178.77 € per ton of wastewater are saved.

$$200 \frac{\text{€}}{t \text{ ww}} - 21.23 \frac{\text{€}}{t \text{ ww}} = 178.77 \frac{\text{€}}{t \text{ ww}}$$

Supplementary Information: carbon footprint estimation:

We aim to compare the introduced ePhoam process with the current treatment of wastewater, which uses the incineration process. For this carbon footprint estimation of wastewater treatment, we consider direct and indirect carbon emissions caused by electricity production off site. As there is little difference in inputs and by-products, their contributions were considered neglectable. The calculation was performed for a production site in Germany in 2023. The emissions are calculated per m³ wastewater. The data and assumption are based on process and energy data provided by Evonik in August 2023. Emission factors were obtained from EU Regulation 601/2012 on the monitoring and reporting of GHG emissions²

❖ Using ePhoam process through an up-scaled roller setup (5 L / 2 hours)

- The Roller treats $5 * 10^{-3} m^3$ of wastewater (ww) for 2 hours.

$$\frac{2 h}{5 * 10^{-3} m^3 \text{ ww}} = 400 \frac{h}{m^3 \text{ ww}}$$

- The Rollers power demand is calculated by multiplying current and voltage.

$$20 V * 20 A = 400 W = 0.4 kW$$

- By multiplying the power demand with the time per m³ wastewater the energy needed for one cubic meter of wastewater is calculated.

$$0.4 kW * 400 \frac{h}{m^3 \text{ ww}} = 160 \frac{kWh}{m^3 \text{ ww}}$$

- The motor that turns the roller has a power demand of 0.18 kW. That multiplied by the time the process needs for cubic meter of wastewater calculates the energy used by the motor per cubic meter wastewater.

$$0.18 \text{ kW} * 400 \frac{\text{h}}{\text{m}^3 \text{ ww}} = 72 \frac{\text{kWh}}{\text{m}^3 \text{ ww}}$$

- The sum of the energy used by the process is $232 \frac{\text{kWh}}{\text{m}^3 \text{ ww}}$.

$$72 \frac{\text{kWh}}{\text{m}^3 \text{ ww}} + 160 \frac{\text{kWh}}{\text{m}^3 \text{ ww}} = 232 \frac{\text{kWh}}{\text{m}^3 \text{ ww}}$$

- The carbon footprint of the process is calculated by multiplying the total energy demand by the CO2 emissions per kWh for Germany which is $0.58 * 10^{-3} \frac{\text{t CO}_2}{\text{kWh}}$.

$$232 \frac{\text{kWh}}{\text{m}^3 \text{ ww}} * 0.58 * 10^{-3} \frac{\text{t CO}_2}{\text{kWh}} = 0.13456 \frac{\text{t CO}_2}{\text{m}^3 \text{ ww}}$$

❖ Using the incineration process

- For incinerating one m^3 wastewater with a low particle content 240 Nm^3 of methane are used.

- With a conversion of $10.55 \frac{\text{kWh}}{\text{Nm}^3 \text{ methane}}$ that equal $5,497 \frac{\text{MWh}}{\text{m}^3 \text{ ww}}$.

$$240 \frac{\text{Nm}^3 \text{ methane}}{\text{m}^3 \text{ ww}} * 10.55 \frac{\text{kWh}}{\text{Nm}^3 \text{ methane}} = 2,532 \frac{\text{kWh}}{\text{m}^3 \text{ ww}} = 2,532 \frac{\text{MWh}}{\text{m}^3 \text{ ww}}$$

- The incineration of the plastic itself produces $0.2 \frac{\text{t CO}_2 \text{e}}{\text{MWh}}$.

$$0.2 \frac{\text{t CO}_2 \text{e}}{\text{MWh}} * 2,532 \frac{\text{MWh}}{\text{m}^3 \text{ ww}} = 0.5064 \frac{\text{t CO}_2 \text{e}}{\text{m}^3 \text{ ww}}$$

- For the raw material of the natural gas the emission factor is $0.46 \frac{\text{t CO}_2 \text{e}}{\text{t methane}}$ and the density of methane is $0.718 \frac{\text{kg}}{\text{m}^3} = 0.718 * 10^{-3} \frac{\text{t}}{\text{m}^3}$.

$$0.46 \frac{\text{t CO}_2 \text{e}}{\text{t methane}} * 240 \frac{\text{Nm}^3 \text{ methane}}{\text{m}^3 \text{ ww}} * 0.718 * 10^{-3} \frac{\text{t methane}}{\text{Nm}^3 \text{ methane}} = 0.0793 \frac{\text{t CO}_2 \text{e}}{\text{m}^3 \text{ ww}}$$

- This sums up to a carbon footprint of $0.5857 \frac{\text{t CO}_2 \text{e}}{\text{m}^3 \text{ ww}}$.

$$0.5064 \frac{\text{t CO}_2 \text{e}}{\text{m}^3 \text{ ww}} + 0.0793 \frac{\text{t CO}_2 \text{e}}{\text{m}^3 \text{ ww}} = 0.5857 \frac{\text{t CO}_2 \text{e}}{\text{m}^3 \text{ ww}}$$

Compared to the new process, $0.45114 \frac{\text{t CO}_2}{\text{t ww}}$ would be saved.

$$0.5857 \frac{\text{t CO}_2}{\text{m}^3 \text{ ww}} - 0.13456 \frac{\text{t CO}_2}{\text{m}^3 \text{ ww}} = 0.45114 \frac{\text{t CO}_2}{\text{t ww}}$$

Using our new cleaning and recovery process, we estimate a carbon footprint reduction by almost **77 %**, without considering the reuse of the recovered particles.

Figure S17. Comparison of cost calculation and carbon footprint estimation between the current incineration process and the ePhoam process using the upscaled Roller setup described in Figure 7 of the main text.

4. XPS fine spectra of elements is required for analyzing the possible chemical change of plastics.

We thank the reviewer for this remark. It is indeed important to analyze the fine spectra of XPS to deduce chemical changes. In the initial revisions, we actually included fine spectra showing the C1s and O1s region of PMMA particles before and after separation. We apologize if there was a miscommunication. We have now further analyzed the chemical composition by peak fitting of the individual sum spectra and have provided these data as a new table in supporting information. These data show no significant difference in the spectra, indicative that the surface of the particles remains essentially unaltered. Importantly, there are no signs of oxidation processes, which would result in increases in the carbonyl peak. The spectra further reveals that the material contains slightly more traces of water after the separation, which could be removed by further drying if needed. It is also important to note that in one of the most important targeted application of PMMA wastewater treatment, the purpose of the process is to enable recycling of particles that would otherwise go to waste. In this case, the product is not used as particles, but will be extruded and form bulk PMMA material (Plexiglass sheets and their likes). Therefore, the key requirement is that the bulk properties remain similar, which we already included via GPC, DSC, and IR measurements (see Figure S13-S15).

Table S8. Results of the curve decompositions of the sum signal for the identification of the C-species and of the O-species

Species (rough assignment)	PMMA before separation		PMMA after separation	
	Binding energy (eV)	Relative. Percentage (%)	Binding energy (eV)	Relative. Percentage (%)
C=O	532.6	42.2	531.6	44.4
C-OH	533.2	55.8	533.1	55.0
Residual moisture (H ₂ O)	535.1	2.0	534.9	0.6
C-C	284.7	55	284.6	59
C-O	286.2	27	286.1	24
O=C-O	288.6	18	288.5	17

Changes to manuscript:

P13: To further confirm this recycling potential, we investigated the thermal properties, molecular weight distributions, chemical composition, as well as surface properties of the PMMA powder before and after separation (Figure S13-S16, Table S7, Table S8). All measurements indicate that the separation process does not degrade or chemically alter the polymers, and thus indicate the possibility to recycle the recovered material in the production process for PMMA products.

Reviewer #4 (Remarks to the Author):

The authors tried hard to reply all my comments though the inherent defects of this study cannot be well overcome or addressed. As a combination of electro-flotation and froth flotation, the proposed technology was not novel from the viewpoint of either technical advances or process mechanisms. There have been a number of similar studies published in the scientific journals in the domain of environmental science and ecology. The paper might be accepted, but if not, I would suggest it to be transferred to Scientific Reports or other journals focusing on separation technologies such as Separation and Purification Technology.

We appreciate the efforts and suggestions of Rev. 4 but politely disagree with her/his interpretation. We believe that our process is innovative precisely because it combines two previously separated fields – particle-stabilized foams and electrophoretic deposition – and translates this combination into a completely new field of application (nanoplastic separation). We further note that there are fundamental differences to electroflotation that we have described in the manuscript and reiterated in reply 1 to reviewer 3. Finally, in our view, our process tackles an unmet challenge. To date, efficient and scalable ways to separate hydrophilic particles that are colloidally stable and remain dispersed in water without agglomeration, do not exist. We have highlighted these differences more clearly in the manuscript:

Abstract:

Here, we describe a process to remove nanoplastics from wastewater, which synergistically combines electrophoretic deposition processes and the formation of stable, particle-stabilized foams enabled by local changes in particle hydrophilicity triggered by the water electrolysis reaction at the electrode surface [...] Both model systems and industrial wastewater were efficiently treated and cleaned from nanoparticles with removal efficiencies exceeding 90%. The exploitation of local pH changes to greatly enhance particle absorption to nascent bubbles directly at the vicinity of the electrode distinguishes our process from conventional electroflotation and allows separation of colloidally stable nanoparticles from aqueous dispersions.

REVIEWERS' COMMENTS

Reviewer #3 (Remarks to the Author):

Authors did not take further in-depth investigations to respond to my main concerns (questions 1 and 2), these two issues are critical for mechanism understanding of the nanoplastic removal process. Hence, I cannot accept this manuscript in its current status.